# Modulating glycosphingolipid metabolism and autophagy improves outcomes in pre-clinical models of myeloma bone disease

Houfu Leng [1], Hanlin Zhang [1], Linsen Li[2], Shuhao Zhang [1,3], Yanping Wang[4], Selina J. Chavda[5], Daria Galas-Filipowicz [5], Hantao Lou[6], Adel Ersek[7], Emma V. Morris [8], Erdinc Sezgin[9,10], Yi-Hsuan Lee[1,7], Yunsen Li[4], Ana Victoria Lechuga-Vieco [1], Mei Tian[11], Jian-Qing Mi[12], Kwee Yong [5], Qing Zhong [2], Claire M. Edwards [8,13], Anna Katharina Simon [1,14] ✉ & Nicole J. Horwood [1,7,14] ✉

Patients with multiple myeloma, an incurable malignancy of plasma cells, frequently develop osteolytic bone lesions that severely impact quality of life and clinical outcomes. Eliglustat, a U.S. Food and Drug Administration-approved glucosylceramide synthase inhibitor, reduced osteoclast-driven bone loss in preclinical in vivo models of myeloma. In combination with zoledronic acid, a bisphosphonate that treats myeloma bone disease, eliglustat provided further protection from bone loss. Autophagic degradation of TRAF3, a key step for osteoclast differentiation, was inhibited by eliglustat as evidenced by TRAF3 lysosomal and cytoplasmic accumulation. Eliglustat blocked autophagy by altering glycosphingolipid composition whilst restoration of missing glycosphingolipids rescued autophagy markers and TRAF3 degradation thus restoring osteoclastogenesis in bone marrow cells from myeloma patients. This work delineates both the mechanism by which glucosylceramide synthase inhibition prevents autophagic degradation of TRAF3 to reduce osteoclastogenesis as well as highlighting the clinical translational potential of eliglustat for the treatment of myeloma bone disease.

Multiple Myeloma (MM) is a hematological cancer caused by abnormal plasma cell expansion in the bone marrow (BM)[1]. MM is preceded by the monoclonal gammopathy of undetermined significance (MGUS); characterized by an abnormal increase in monoclonal immunoglobulin secretion[2,3]. Progression of the disease from MGUS into active MM is accompanied by the development of osteolytic bone disease[4], affecting 85% of MM patients[5].

MM cells secrete factors, including receptor activator of nuclear factor kappa-B ligand (RANKL), that enhance osteoclast (OC) formation[6]. Engagement of RANKL, the OC differentiation factor, with its cognate receptor RANK leads to the activation of the NF-κB signaling pathway and is required for OC formation. The NF-κB pathway is regulated by tumor necrosis factor receptor-associated factors (TRAFs). In particular TRAF6 mediates RANKL-induced osteoclastogenesis[7] whilst TRAF3 is inhibitory[8]. Treatment for MM bone disease involves blocking OC formation and/or activity with agents such as bisphosphonates or the monoclonal antibody denosumab that directly blocks RANKL[9]; similar efficacy at preventing bone destruction has been shown with either treatment[10]. The bisphosphonate, zoledronic acid (ZA), is a frontline treatment for myeloma bone disease; however, for some individuals ZA treatment may not be well tolerated or may lead to complications[11,12]. Thus, there is an unmet clinical need for alternative ways to prevent the progression, pain, and disability of myeloma bone disease.

Autophagy maintains eukaryotic cell homeostasis by degrading and recycling cellular waste including protein aggregates and damaged organelles[13]. The autophagy process occurs in a stepwise manner: (1) initiation of membrane isolation; (2) nucleation; (3) elongation to form an autophagosome; and (4) fusion to the lysosome to form an autolysosome. Specifically, induction is mediated by phosphorylation of UNC-51-like kinase 1 (ULK1) complex that triggers the translocation of Class III PI3K complex containing beclin-1, Vps34, AMBRA, ATG14L, and p115 to initiate phagophore nucleation[13,14]. The ATG12 conjugation complex including ATG7 and ATG10 which mediate the conjugation of ATG12 to ATG5 are required during autophagosome elongation and cargo sequestration[13,14]. Nascent LC3 is processed by ATG4, ATG7, and ATG3, which ultimately forms the conjugated LC3-phosphatidylethanolamine (PE) (LC3-II); an autophagy marker frequently used to monitor autophagic flux[13].

During OC differentiation Beclin-1 is upregulated[15] and Atg7 knockdown inhibits the expression of key OC proteins such as tartrate-resistant acid phosphatase (TRAP) and cathepsin K[16]. Additionally, autophagy-related proteins including ATG5, ATG7, ATG4B, and LC3 regulate a number of OC functions[17]. Concordantly, autophagy inhibitors such as chloroquine (CQ) can inhibit OC formation and restore bone mass[8]. Molecularly, TRAF3, as a suppressor of OC differentiation, is degraded by autophagy upon RANKL induction in BM OC precursors[8]. Specifically, TRAF3, a cytosolic scaffold, represses the activation and nuclear translocation of RELB, a non-canonical NF-κB transcription factor[18]. Binding of TRAF3 to NDP52, an autophagic cargo receptor, causes NDP52-mediated degradation of TRAF3 via autophagy and results RELB activation[19,20]. The co-localization of TRAF3 with LC3B indicates that TRAF3 is targeted to the autophagosome[19], while silencing of NDP52 disrupted the co-localization of TRAF3 with LC3B and elevated endogenous TRAF3 protein level[19]. In addition, treatment with Bafilomycin A1 (BafA1), an autophagy inhibitor, caused co-localization of TRAF3 with the lysosomal marker LAMP2, indicating that TRAF3 reaches the lysosomes[19]. Finally, by using RNAi against ATG5 or against ULK1 in A549 cells to inhibit autophagy, TRAF3 protein levels were increased, inhibiting p100 to p52 processing and blocking nuclear translocation of RELB[19]. Together these findings suggest that TRAF3 is degraded by the autolysosomal pathway. Based on these findings, it has been proposed that autophagy inhibitors may be beneficial in treating bone loss diseases in the clinic.

Glycosphingolipids (GSLs) are normal constituents of the cell membrane that are expressed in varying ratios and combinations. We previously observed that GM3, a ganglioside abnormally expressed in MM cells, promotes osteoclastogenesis and that this could be blocked using miglustat[21], an iminosugar and an analogue of D-glucose[22]. However, miglustat can act on a range of biological and pathological pathways leading to deleterious side effects in the gastrointestinal tract including severe diarrhea and weight loss[23]. Eliglustat, a small synthetic molecule, specifically prevents the synthesis of all GSLs by inhibition of glucosylceramide synthase (GCS)[24] and is used for the treatment of Gaucher disease type 1 in adults[25]. Gaucher patients suffer from osteoporosis/osteopenia and treatment to reduce excess bioactive GSL associated with the disease has been shown to alleviate bone loss[26]. Interestingly, patients with Gaucher disease have a 6–50 times increased risk of developing MM or the pre-MM MGUS condition[27–29]. However, it remains to be determined whether the reduction in bone symptoms is due to the treatment of Gaucher disease or if eliglustat acts directly on bone cells, which raised the possibility that eliglustat may be of benefit in treating MM bone complications.

Here we demonstrate that eliglustat reduces bone disease by directly acting on bone cells, namely OCs, in preclinical MM and MM-like MGUS models and can be used in combination with ZA for superior outcomes. Mechanistically, eliglustat acts as an autophagy inhibitor that prevents TRAF3 degradation in OC and modulates autophagy by altering glucosylceramide (GlcCer) and lactosylceramide (LacCer) composition in OC. Lastly, using MM patient bone marrow samples, eliglustat effectively inhibited OC formation while replacement of the GSLs rescued OC formation.

## Results

### Eliglustat increases trabecular bone volume by inhibiting OCs in healthy mice

To determine the effect of eliglustat treatment on normal bone mass, female C57BL/6 J mice were fed with normal chow or eliglustat-containing chow (150 mg/kg/day for all in vivo experiments[29]) for 19 days prior to collecting the tibiae from the mice for micro-CT analysis−19 days was selected to coincide with the treatment regime used in the MM models. Eliglustat treatment improved trabecular bone as evidenced by micro-CT reconstruction images (Fig. 1a). Quantification demonstrated a significant increase in bone parameters including bone volume over total volume (BV/TV), bone surface (BS), trabecular number (Tb.N), and connective density (Conn.D), and a decrease in bone surface density (BS/BV) and in trabecular separation (Tb.Sp) (Fig. 1b–g). Both BS and BV increased in the eliglustat group; however, as the BV increase was greater than BS, there was an overall decrease of BS/BV ratio. Experiments in male C57BL/6J mice similarly showed that there was an increase in bone indices in the presence of eliglustat (Supplementary Fig. 1a–g). The increase in trabecular bone was not accompanied by a change in either body weight or bone marrow adipose tissue (BMAT) volume, indicating that eliglustat does not substantially affect adipocyte development (Supplementary Fig. 1h–i).

To determine whether bone-resorbing OCs or bone-forming osteoblasts (OBs) were the target of eliglustat, their numbers in the trabecular bone were quantified by histomorphometric analysis. Longitudinal cross-sections of tibiae were stained with TRAP to indicate OC. OBs were identified by their typical cuboidal morphology along the bone surface (Fig. 1h). Eliglustat-treated mice tibiae showed significantly lower measurements for OC surface, number and perimeter when compared with naïve control mice (Fig. 1i–k). However, no difference was observed between untreated control mice and eliglustat-treated mice for OB parameters (Fig. 1l–n), indicating that the observed increases in trabecular bone were a consequence of OC inhibition rather than an increase in OB formation.

### Eliglustat inhibits osteoclastogenesis leading to a decrease MM bone disease in vivo

To investigate the effects of eliglustat in MM bone disease model, female C57BL/KaLwRijHsd mice were injected with saline or 5TGM1-GFP murine MM cells on day 0. Eliglustat-containing chow was given from day 4 until sacrifice on day 23. Micro-CT reconstruction of tibiae indicated a decrease in trabecular bone in MM-bearing mice compared to saline group, whilst the eliglustat group was protected from MM-induced bone loss (Fig. 2a). Treatment in MM-bearing mice significantly increased BV/TV, BS, and Tb.N, accompanied by a decrease in BS/BV and in Tb.Sp (Fig. 2b–h), confirming an inhibition of bone catabolism. Similarly, eliglustat treatment in MM-bearing male mice was able to improve MM bone indices (Supplementary Fig. 2a–h). Importantly, treatment with eliglustat significantly reduced the number of cortical bone lesions in MM-bearing mice (Fig. 2i–j).

Histomorphometric analysis demonstrated a significant increase in TRAP-positive OCs on the endosteal surface in MM-bearing mice compared to control mice (Fig. 2k). Treatment of MM mice with eliglustat exhibited a robust response as shown by decreased OC parameters including surface, number and perimeter (Fig. 2l–n); in concordance with OC results obtained with healthy C57BL/6J mice fed on an eliglustat-containing diet (Fig. 1i–k). Eliglustat treatment did not

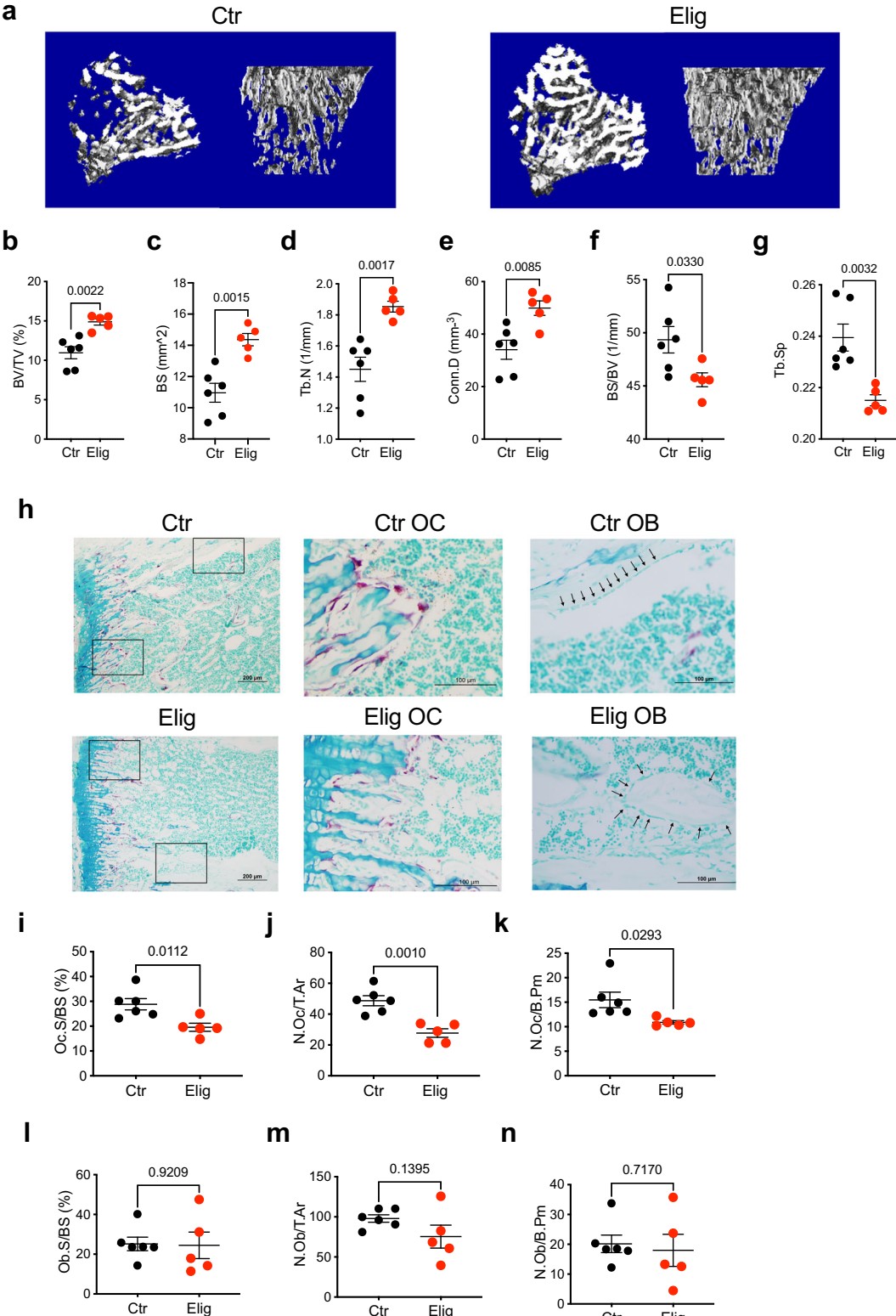

**Fig. 1 | Eliglustat increases trabecular bone in healthy mice by inhibiting OC in vivo. a** Representative micro-CT reconstruction images of tibiae from 8-week-old female C57BL/6J mice treated with normal chow (Ctr, $n = 6$ biologically independent animals) or eliglustat (Elig, $n = 5$ biologically independent animals) chow for 19 days. **b**–**g** Micro-CT analysis of tibiae: BV/TV, BS, Tb.N, Conn.D, BS/BV, and Tb.Sp (Ctr, $n = 6$ and Elig, $n = 5$ biologically independent animals). **h** Representative TRAP/0.2% methyl green stained tibial sections showing OCs (red/purple) and OBs (black arrowhead) on the endocortical bone surface from each group (left); magnified boxed areas in images on the right. Result is representative of two independent experiments. **i**–**k** Bone histomorphometry quantification of OCs includes the percentage of OC surface over total bone surface (Oc.S/BS), number of OCs over total bone area (N.Oc/T.Ar), and number of OCs per bone perimeter (N.Oc/B.Pm) (Ctr, $n = 6$ and Elig, $n = 5$ biologically independent animals). **l**–**n** Bone histomorphometry quantification of OB including percentage of OB surface over total bone surface (Ob.S/BS), number of OBs over total bone area (N.Ob/T.Ar) and number of OBs per bone perimeter (N.Ob/B.Pm). Ctr, $n = 6$ and Elig, $n = 5$ biologically independent animals, each dot is one animal. Data are presented as mean values ± SEM. Exact $p$-values are depicted in the figure. Statistical analysis was performed using unpaired two-tailed Student's $t$-test for **b**–**g** and **i**–**n**. Source data are provided as a Source Data file.

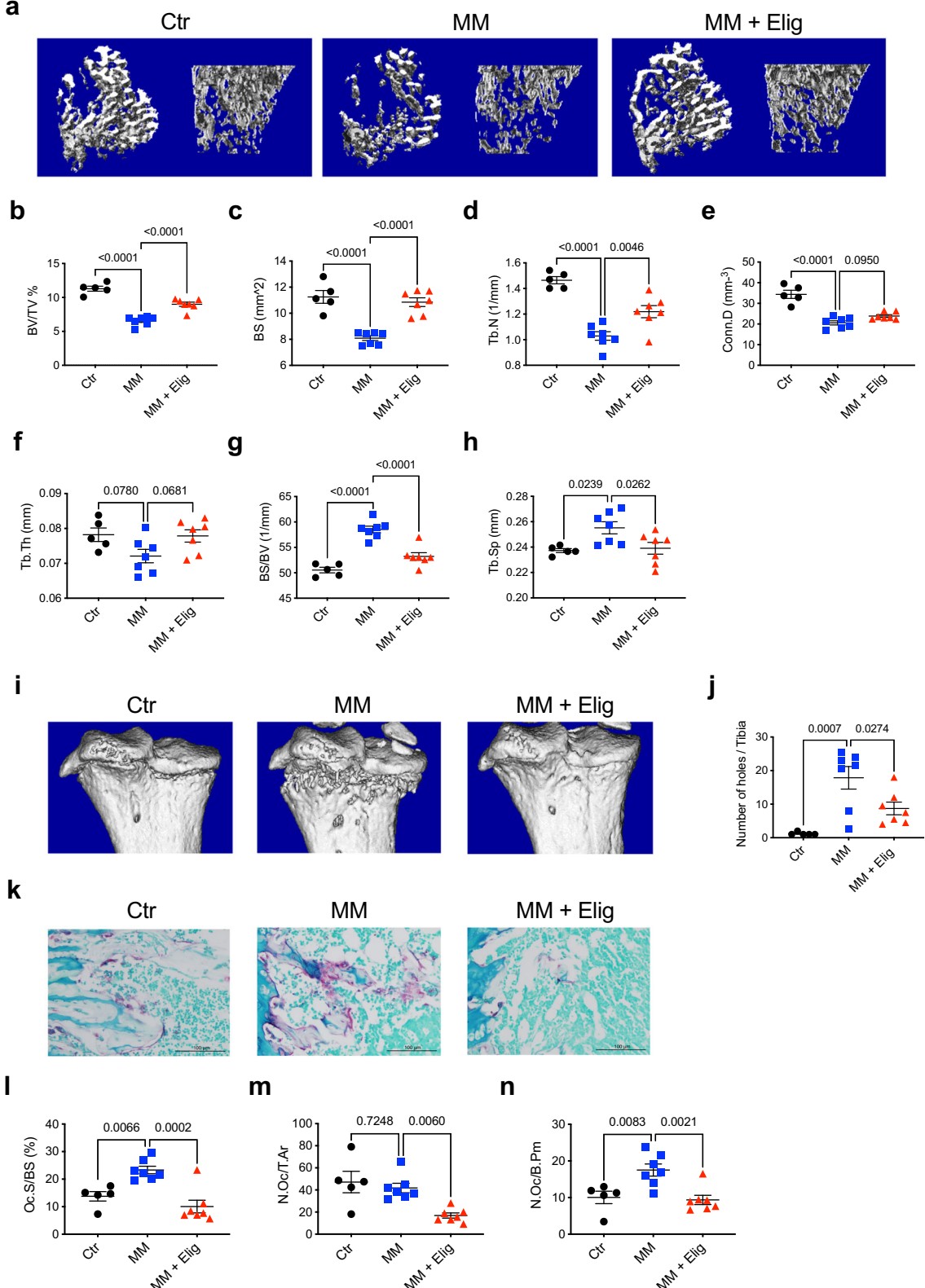

**Fig. 2 | Eliglustat ameliorates 5TGM1-GFP MM cell-induced bone disease.** 5TGM1-GFP MM cells were injected to 8-week-old female C57BL/KaLwRijHsd mice to generate the MM model. Eliglustat chow was administered from day 4 post tumor injection (until day 23). **a** Representative micro-CT reconstruction images of tibiae trabecular bones from naive control (Ctr, $n = 5$ biologically independent animals), MM mice with normal chow (MM, $n = 7$ biologically independent animals), or MM mice with eliglustat chow (MM + Elig, $n = 7$ biologically independent animals). **b**–**h** Tibiae trabecular bone parameters were assessed: BV/TV, BS, Tb.N, Conn.D, Tb.Th, BS/BV, and Tb.Sp (Ctr, $n = 5$; MM, $n = 7$, and MM + Elig, $n = 7$

biologically independent animals). **i, j** Tibiae cortical bone reconstruction (**i**) and the number of cortical bone lesions (**j**) (Ctr, $n = 5$; MM, $n = 7$, and MM + Elig, $n = 7$ biologically independent animals). **k**–**n** Representative TRAP/0.2% methyl green staining showing red OCs of tibial histological sections with original magnification ×40 (**k**) and the quantification of OCs with Oc.S/BS, N.Oc/T.Ar, and N.Oc/B.Pm (**l**–**n**) (Ctr, $n = 5$; MM, $n = 7$, and MM + Elig, $n = 7$ biologically independent animals). The result is representative of two independent experiments. Data are presented as mean values ± SEM. Exact $p$-values are depicted in the figure. Statistical analysis was performed using One-way ANOVA. Source data are provided as a Source Data file.

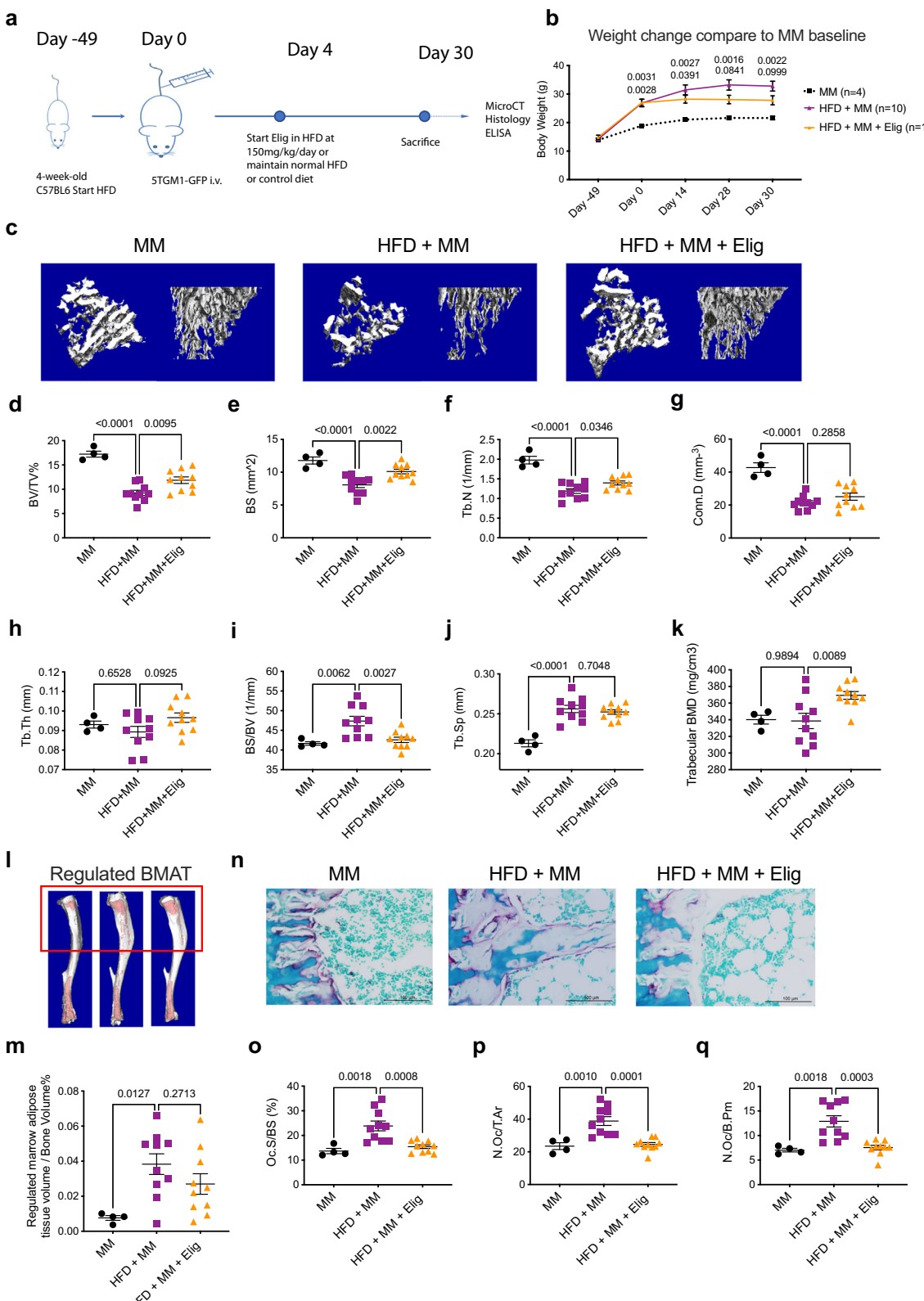

change the population of OC precursors as demonstrated by expression of surface markers (CD11b−CD3−B220−CD115+)[30] in MM-bearing mice suggesting that eliglustat blocks OC formation rather than inhibiting the generation of OC progenitors (Supplementary Fig. 2i–k).

Flow cytometry analysis of BM and spleen showed that there was no difference in the percentage of GFP + 5TGM1 cells found in

BM (Supplementary Fig. 2l) or spleen (Supplementary Fig. 2m) between MM-bearing mice and mice treated with eliglustat (Supplementary Fig. 2n–o). Consistently, serum paraprotein IgG2bκ showed that eliglustat did not reduce systemic tumor burden (Supplementary Fig. 2p). Therefore, eliglustat ameliorates MM-induced bone disease by specifically inhibiting OC without exacerbating tumor burden.

**Fig. 3 | Eliglustat reduces bone loss in a diet-induced obesity MGUS model.** **a** Schematic overview and experiment design. Four-week-old female C57BL/6J mice were divided into MM injection with normal diet (MM, $n = 4$ biologically independent animals), HFD with MM (HFD + MM, $n = 10$ biologically independent animals), and HFD with MM plus eliglustat (HFD + MM + Elig, $n = 10$ biologically independent animals). **b** Body weight (g) was measured over the duration of the experiment. Statistical analysis was performed by comparing with MM group (MM, $n = 4$; HFD + MM, $n = 10$, and HFD + MM + Elig, $n = 10$ biologically independent animals). **c** Representative micro-CT reconstruction images of tibiae trabecular bones from respective groups. **d–k** Micro-CT analysis parameters of tibiae: BV/TV, BS, Tb.N, Conn.D, Tb.Th, BS/BV, Tb.Sp, and trabecular BMD (MM, $n = 4$; HFD + MM, $n = 10$, and HFD + MM + Elig, $n = 10$ biologically independent animals). **l, m** Osmium tetroxide staining to detect regulated BMAT (in red box) by micro-CT in MM, HFD + MM, and HFD + MM + Elig groups (MM, $n = 4$; HFD + MM, $n = 10$, and HFD + MM + Elig, $n = 10$ biologically independent animals). **n–q** Representative TRAP/0.2% methyl green stained tibial histological sections showing red OCs (**n**). The result is representative of two independent experiments. Bone histomorphometry parameters including Oc.S/BS (**o**), N.Oc/T.Ar (**p**), and N.Oc/B.Pm (**q**) (MM, $n = 4$; HFD + MM, $n = 10$, and HFD + MM + Elig, $n = 10$ biologically independent animals). Data are presented as mean values ± SEM. Exact $p$-values are depicted in the figure. Statistical analysis was performed using One-way ANOVA. Source data are provided as a Source Data file.

## Eliglustat prevents bone loss in a diet-induced obesity model of MGUS

MGUS is a common occurrence in the elderly and carries a risk of progression to MM at ~1% per year. MGUS patients do not present with lytic bone lesions or hypercalcemia, but they do have a greater risk of developing osteoporosis and increased fracture incidence[31]. When fed standard laboratory chow, C57BL/6J mice are refractory to the engraftment of 5TGM1 cells, as opposed to the MM susceptible C57BL/KaLwRijHsd mice as shown in Fig. 2. However, research has shown that altering the mouse chow to a high fat diet (HFD) creates a permissive environment for MM cell engraftment in C57BL/6J mice and the development of an MGUS-like phenotype characterized by elevated serum paraprotein and mild changes in bone measurements[2].

To expand the potential application of eliglustat, C57BL/6J mice were fed with either 42% HFD or control diet for 7 weeks, resulting in a significant increase in body weight for the HFD group (Fig. 3a, b). Serum paraprotein measurements taken at sacrifice confirmed that C57BL/6J mice on a control diet inoculated with 5TGM1 cells (MM group) showed no sign of tumor growth as previously reported[2] whilst mice in the HFD + MM group showed elevated serum paraprotein levels; treatment with eliglustat did not reduce serum paraprotein (Supplementary Fig. 3a). 7 weeks of HFD alone statistically increased body weight (Supplementary Fig. 3b) but did not change any of the bone parameters measured (Supplementary Fig. 3c–h). However, when investigating the HFD + MM group, micro-CT reconstruction showed decreased tibial trabecular bone mass in these mice with the MGUS-like condition compared to the mice placed on normal diet with MM cells (Fig. 3c). As expected based on the earlier figures, eliglustat reversed bone loss with a significant increase in BV/TV, BS, Tb.N, and trabecular bone mineral density (BMD), together with a decreased BS/BV ratio (Fig. 3d–k).

The bone marrow microenvironment plays an important role in MM progression and bone disease. With age and obesity, there is a significant increase in the proportion of bone marrow adipocytes; these cells provide an energy store, adipokines and bioactive substances, that MM cells utilize for migration, growth and survival[32]. BMAT is also supportive of OC formation[33]. To determine whether eliglustat impacted OC formation by suppressing HFD-induced BMAT, tibiae were stained with osmium tetroxide; a heavy metal that binds to lipids allowing imaging via micro-CT[34]. Quantitative analysis of regulated BMAT (rBMAT, red area within the red rectangle) revealed a significant increase of rBMAT in HFD + MM (MGUS) compared to the MM group (control); eliglustat treatment did not alter rBMAT volume indicating that the effects of eliglustat were not due to changes in adiposity in the BM (Fig. 3l, m). OC measurements were increased in HFD + MM (MGUS) group as compared to MM group (control) and treatment with eliglustat significantly reversed the pattern (Fig. 3n–q) without affecting OB parameters or cortical bone parameters (Supplementary Fig. 3i–o).

## Eliglustat combined with ZA has a superior effect compared to ZA alone

ZA leads to OC apoptosis via blockade of the mevalonate pathway[35], while our data show that eliglustat inhibits OC via an as yet unknown mechanism. The efficacy of combining these two OC inhibitors was evaluated to ascertain any potential additive effect that would also allow ZA to be used at lower concentrations to mitigate the potential to develop osteonecrosis of the jaw (ONJ).

C57BL/KaLwRijHsd mice were divided into five groups: untreated control mice (Ctr); 5TGM1-GFP MM-bearing mice (MM); MM-bearing mice treated with ZA (MM + ZA); MM-bearing mice treated with eliglustat (MM + Elig); or MM-bearing mice treated with both ZA and eliglustat (MM + ZA + Elig) (Fig. 4a). Treatment with the combination of eliglustat and ZA resulted in a two-fold increase in BV/TV compared to MM-bearing mice and a 1.5-fold increase when compared with mice treated with eliglustat alone or ZA alone (Fig. 4b, c). Importantly, the combination strategy increased BS and Tb.N when compared with ZA given alone, whilst BS/BV was significantly decreased (Fig. 4d–h). Consistently, the combination strategy significantly decreased the number of osteolytic lesions (holes) compared to single use of eliglustat or ZA (Fig. 4i–j). Histomorphometric analysis revealed that the combination strategy was superior to single use of each compound in inhibiting OC surface (Fig. 4k–n). Neither eliglustat nor ZA demonstrated a significant effect on OB parameters (Supplementary Fig. 4a–c). Thus, the combination of eliglustat and ZA may provide a better clinical strategy than ZA alone for MM patients.

Overall, treatment with eliglustat-inhibited osteoclastogenesis in healthy mice, MM susceptible mice and the HFD-induced MGUS murine model. Additionally, eliglustat combined with ZA produced an even greater bone-sparing effect. To elucidate how eliglustat inhibits OC, further investigation of the molecular mechanisms was undertaken.

## OC inhibition with eliglustat depends on TRAF3

RAW264.7 cells differentiated with M-CSF and RANKL in vitro were used as OC precursors in mechanistic experiments due to the ability to obtain high numbers of OC without other contaminating cell types in the BM. Using a dose range below or equivalent to that used clinically (50 μM)[29], eliglustat-inhibited TRAP-positive OC formation in a dose-dependent manner (Fig. 5a), as quantified by OC number, nuclei number per OC and OC area (Fig. 5b–d). In contrast, OC precursor viability was not affected (Fig. 5e). Similarly, eliglustat inhibited OC formation from murine BM cells (Supplementary Fig. 5a).

RANKL triggers OC formation via the canonical and non-canonical NF-κB pathways (Fig. 5f), both essential for OC formation. In the canonical NF-κB pathway, activation of the adaptor protein TRAF6 leads to proteasome-mediated IκBα degradation followed by nuclear translocation of p65 and p50[36]; in the non-canonical NF-κB pathway, the adaptor protein TRAF3 is degraded in an autophagosome/lysosome-dependent mechanism[8], that induces nuclear translocation of p52 and RelB[37]. To determine whether the eliglustat-induced decrease in OC was related to TRAF3 degradation, TRAF3 levels were measured after one day of treatment with RANKL and M-CSF in the presence or absence of either eliglustat or the autophagic flux inhibitor BafA1. As expected, inhibition of autophagy with BafA1 led to the accumulation of TRAF3 protein in RANKL-stimulated RAW264.7 cells and in primary murine BM cell-derived OCs, whereas *traf3* mRNA was not affected (Fig. 5g–i and Supplementary Fig. 5b–d). In the presence of eliglustat there was a similar accumulation of TRAF3 indicating that eliglustat

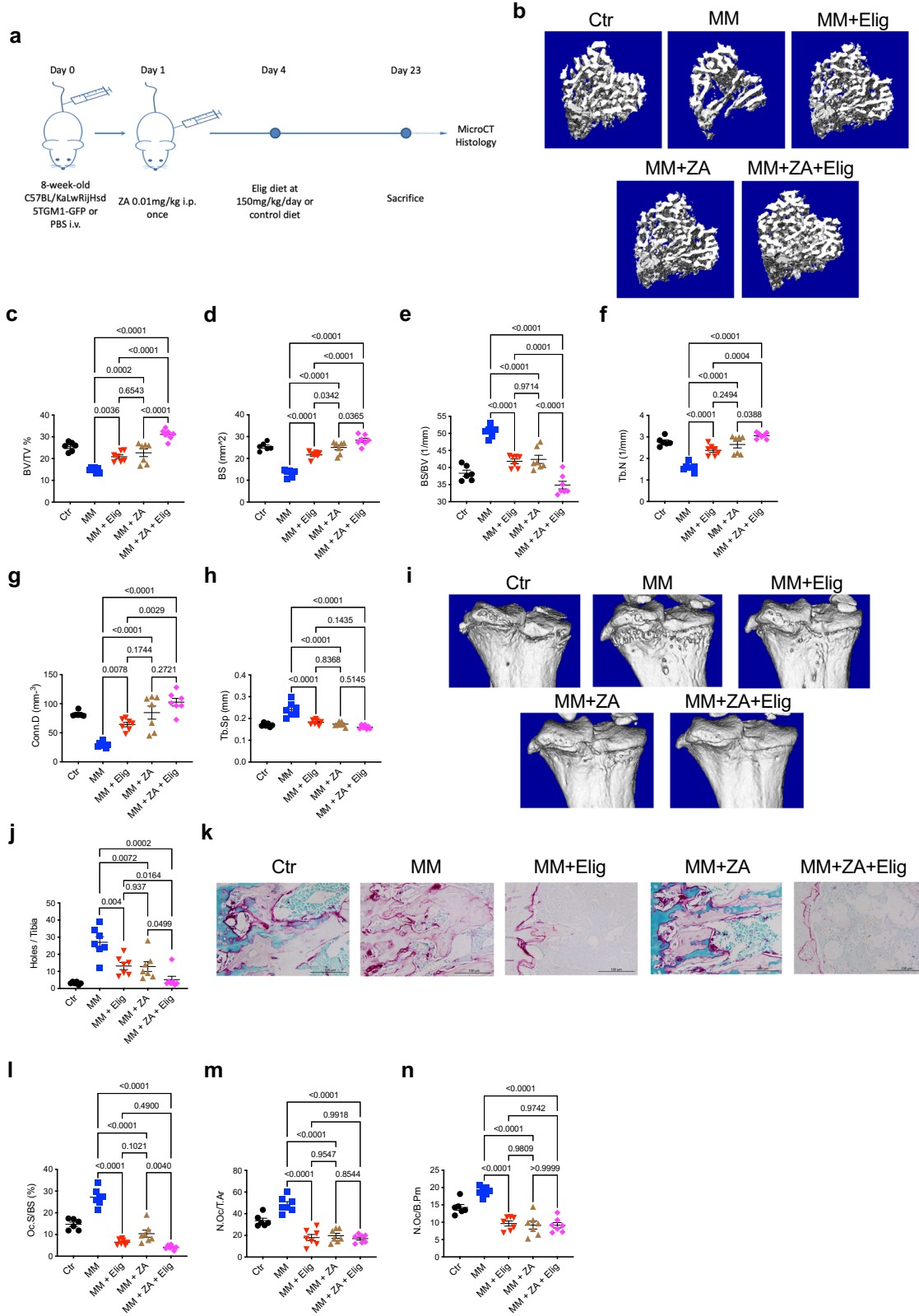

blocks TRAF3 lysosomal degradation elicited by RANK activation (Fig. 5g–i and Supplementary Fig. 5b–d).

For the canonical NF-κB pathway, we next tested whether eliglustat affects proteasome degradation, an essential step in the canonical TRAF6/NF-κB pathway. The proteasome is known to degrade IκBα which allows RANKL-mediated canonical NF-κB signaling cascades to occur that are essential for OC formation (Fig. 5f)[36,38–40].

The proteasome–ubiquitin inhibitor MG132 was added as a positive control to this series of experiments. As expected, RANKL upregulated TRAF6 in murine primary BM cells (Supplementary Fig. 5e)[41]. As previously published, BafA1[42,43] and MG132[44,45] did not affect TRAF6 levels. TRAF6 also remained elevated in the presence of eliglustat (Fig. 5j–k). Next, our data demonstrate that MG132 prevented IκBα degradation as expected[8]; however, they were not affected by either eliglustat or BafA1

**Fig. 4 | Eliglustat combined with ZA reduces MM bone disease with greater effect than either agent alone. a** Schematic illustrating the timeline and experimental design of eliglustat and ZA impact on bone measurements in 8-week-old C57BL/KaLwRijHsd 5TGM1-GFP MM-bearing male mice. **b** Representative images of micro-CT reconstruction of proximal tibiae from each group, including naive control (Ctr, $n = 6$ biologically independent animals), MM mice (MM, $n = 7$ biologically independent animals), MM mice with eliglustat chow (for 19 days) (MM + Elig, $n = 7$ biologically independent animals), MM mice with single dose ZA injection (0.01 mg/kg, MM + ZA, $n = 7$ biologically independent animals), or MM mice with single dose ZA injection (0.01 mg/kg), and eliglustat chow (for 19 days) (MM + ZA + Elig, $n = 7$ biologically independent animals). **c–h** Dot plots of BV/TV, BS, BS/BV, Tb.N, Conn.D, and Tb.Sp from each group (Ctr, $n = 6$; MM, $n = 7$; MM + Elig, $n = 7$; MM + ZA, $n = 7$; MM + ZA + Elig, $n = 7$ biologically independent animals). **i** Representative reconstruction images of proximal cortical bone from each group. **j** Bone lesions (holes) on the cortical bones from each mouse were counted (Ctr, $n = 6$; MM, $n = 7$; MM + Elig, $n = 7$; MM + ZA, $n = 7$; MM + ZA + Elig, $n = 7$ biologically independent animals). **k** Representative TRAP/0.2% methyl green stained tibial sections showing red OCs on the endocortical bone surface from each group. The result is representative of two independent experiments. **l–n** Bone histomorphometry parameters including Oc.S/BS, N.Oc/T.Ar, and N.Oc/B.Pm (Ctr, $n = 6$; MM, $n = 7$; MM + Elig, $n = 7$; MM + ZA, $n = 7$; MM + ZA + Elig, $n = 7$ biologically independent animals). Data are presented as mean values ± SEM. Exact $p$-values are depicted in the figure. Statistical analysis was performed using One-way ANOVA for **c–h** and **l–n**, and two-tailed $t$-test for **j**. Source data are provided as a Source Data file.

(Fig. 5l–m). These observations indicate that eliglustat does not regulate the TRAF6-mediated canonical NF-κB pathway and confirms that the TRAF3 regulated non-canonical NF-κB pathway is likely to be its target.

To confirm the role of TRAF3 in mediating eliglustat's OC-inhibitory effects in vivo, BM chimeric mice were generated using BM donor cells from LysM-Cre+, TRAF3fl/fl mice that harbor a myeloid specific deletion of *traf3* (Supplementary Fig. 5f–g)[8]. Chimeric mice were fed an eliglustat-containing diet for 19 days prior to micro-CT analysis. In accordance with Fig. 1, eliglustat administration led to an increase of BV/TV in mice that received wild-type BM donor cells; however this increase was no longer apparent in chimeric mice that received BM from TRAF3-deficient mice (Fig. 5n and Supplementary Fig. 5h–n). In addition, histomorphometric analysis revealed that OC measurements were not affected by eliglustat in chimeric mice that received BM from TRAF3-deficient mice (Supplementary Fig. 5o–q), further confirming that eliglustat suppresses OC formation in a TRAF3-dependent manner.

### Eliglustat is an autophagy inhibitor that prevents autolysosomal degradation

The degradation of TRAF3 in RANKL-induced OC formation is mediated by autophagy machinery[8]. Therefore, whether eliglustat acts as an autophagic inhibitor was investigated. Upon eliglustat treatment, increased accumulation of the autophagosomal and autolysosomal marker LC3-II was observed in RAW264.7 cells; similar to treatment using the autophagic flux inhibitor BafA1. BafA1 did not increase LC3-II levels in eliglustat-treated RAW264.7 cells any further suggesting that the accumulated LC3-II in eliglustat-treated cells was due to autophagic flux inhibition (Fig. 6a, b). p62 is an autophagic receptor that attracts cargo into the autophagosomal lumen. Like LC3-II, it is degraded when the autophagsomal content is delivered to the lysosome. In this study, p62 degradation was inhibited by eliglustat as well, indicating that eliglustat acts as an autophagy inhibitor, most likely at the lysosomal step of the autophagic flux (Fig. 6c, d).

Consistently, eliglustat treatment led to significant accumulation of GFP-LC3 puncta in transfected cell system (Fig. 6e, f)[46]. Moreover, a tandem-tagged RFP-GFP-LC3 reporter system was used to determine whether the accumulated LC3 puncta are on autophagosomes or autolysosomes. In this system, GFP but not RFP is quenched upon autophagic delivery to the acidic lysosome[46]. Eliglustat treatment led to increased yellow puncta and reduced red puncta (Fig. 6g–j), indicating that the accumulation of autophagosomes (or dysfunctional autolysosomes) was associated with reduced lysosomal degradation. Furthermore, transmission electronic microscopy revealed that eliglustat accumulated single-membrane autolysosomes rather than double-membrane autophagosomes (Fig. 6k, l), similar to the effect of CQ and BafA1 treatment.

To further investigate TRAF3 distribution after eliglustat treatment, a lysosome isolation kit was used and demonstrated that eliglustat significantly increased the amount of TRAF3 in the LAMP1+ lysosome-enriched layer (Fig. 6m, n). In addition, it was observed that eliglustat significantly increased the area of the lysosomal marker LAMP2, suggesting an increase of lysosomes and/or autolysosomes (Supplementary Fig. 6a, b). To investigate the effects of eliglustat on trafficking and accumulation of TRAF3 to lysosomes in RAW264.7 cells more precisely, TRAF3 subcellular localization was performed in pre-OCs by confocal microscopy. It was previously reported that TRAF3 colocalizes with lysosome after BafA1 treatment[19] and a similar pattern was observed after eliglustat treatment for 60 min and 120 min as indicated by a significant increase in the TRAF3-LAMP2 co-localization ratio (Fig. 6o, p and Supplementary Fig. 6c, d).

To test if other autophagy inhibitors support this proposed mechanism, four different inhibitors were investigated for their effects on OC formation; 3-methoxyamphetamine (3MA, class I PI3K inhibitor) and SAR405, both of which are VPS34 inhibitors and prevent the nucleation process upstream of autophagy as well as BafA1 and CQ, both lysosomal inhibitors, that disrupt the completion process of autophagy[47,48]. All these four reagents were able to significantly inhibit OC formation consistent with previous published findings for 3MA[49], CQ[8], and BafA1[50] (Supplementary Fig. 6e, f).

Taken together, these data support a role for eliglustat as an autophagy inhibitor that blocks autophagic flux by inhibiting autolysosomal degradation.

### Eliglustat's effect on autophagy is mediated by the inhibition of GSL synthesis

Autophagy is critical to osteoclastogenesis and gangliosides are involved in autophagosomal biogenesis[14,51]. It was hypothesized that eliglustat prevents the conversion of ceramide to certain GSLs that are required for autophagy.

To evaluate the overall lipid changes on the cellular membranes, the lipid organization of eliglustat-treated RAW264.7 cells was evaluated by spectral imaging (Fig. 7a). The generalized polarization (GP) value reveals how packed the membrane lipids are[52]. Eliglustat significantly decreased the GP value (Fig. 7b), indicating a higher fluidity of the plasma membrane. It is possible that overall fluidity including that of endomembranes, such as autophagosome or lysosome membranes, may be affected in a similar way. However, due to technical limitations, it was difficult to specifically measure the fluidity of endomembranes.

To profile the specific GSL composition changes, LTQ-ESI-MS glycosphingolipidomics was applied. Similar to D-PDMP, a known GCS inhibitor[53], eliglustat-treated RAW264.7 cells showed significant reduction of GlcCer, LacCer and overall GSLs (Fig. 7c–f). Full Original Lipidomics Data are available in Supplementary Data 1. To investigate whether exogenous GlcCer or LacCer may be sufficient to rescue eliglustat-induced autophagy deficiency, three commercially available lipids, LacCer (C16 and C24), and GlcCer (C16), were investigated. The LacCer and GlcCer significantly reversed, albeit not fully, the accumulated LC3II protein levels (Fig. 7g, h) and the accumulation of TRAF3 caused by eliglustat (Fig. 7i, j). Hence, by removing certain GSLs from OC, eliglustat inhibits the process of autophagy and maintains TRAF3 protein levels thus inhibiting OC formation.

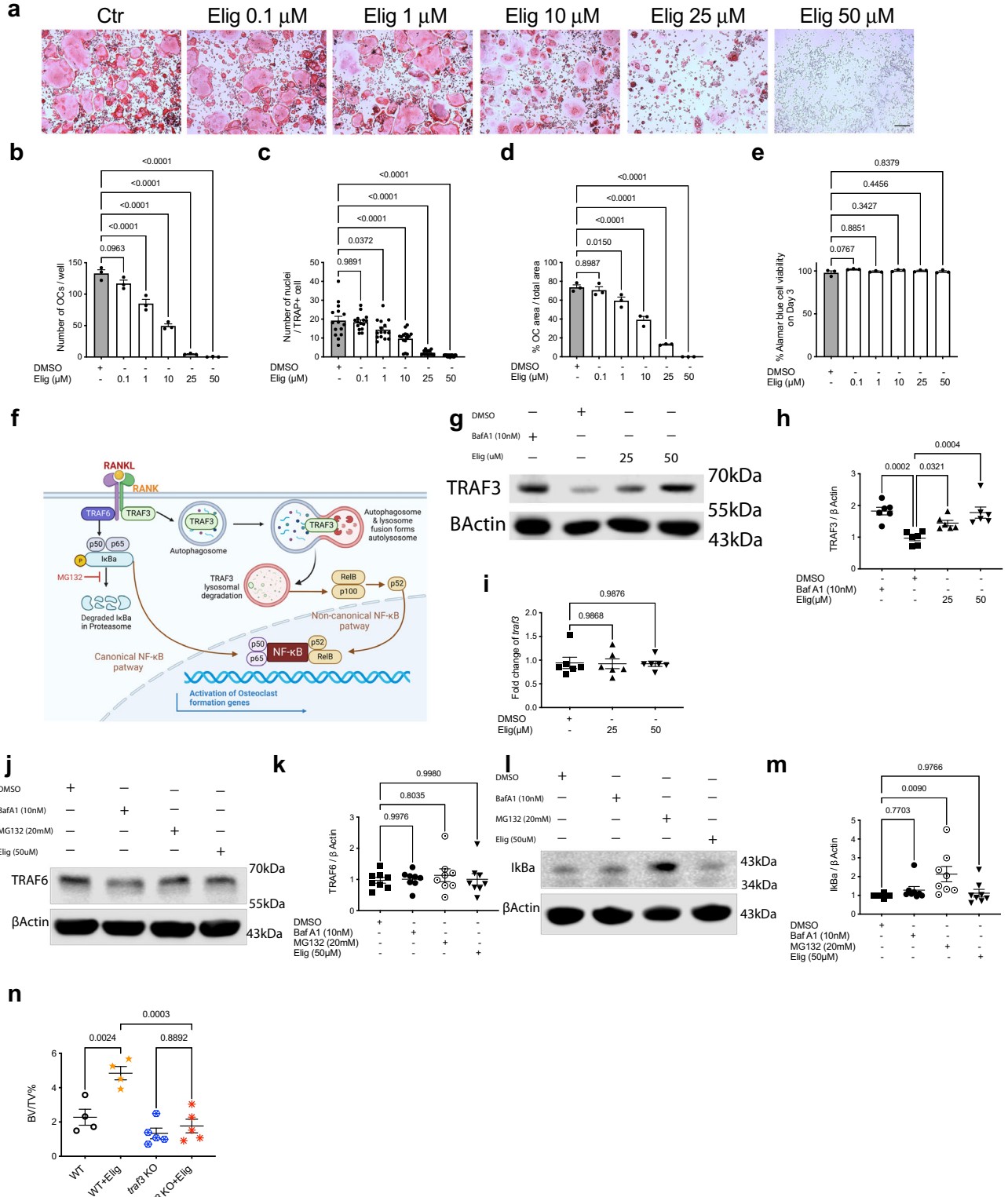

## Eliglustat inhibits OCs derived from MM patients

MM patients frequently suffer from bone lesions resulting from increased OC-mediated bone resorption. To assess the potential clinical efficacy of eliglustat, OCs from MM patient BM mononuclear cells aspirates were treated with eliglustat (Supplementary Table 1). Consistent with our findings in RAW264.7 cells (Fig. 5a–d), eliglustat effectively inhibited primary human OC formation in a dose-dependent manner (Fig. 8a–d). Moreover, eliglustat-inhibited OC formation from MM patients' BM mononuclear cells was reversed by

exogenous LacCer or GlcCer (Fig. 8e–h). These data provide promising evidence that through modulating LacCer and GlcCer levels, eliglustat may be used to inhibit MM patient OC formation for resolving bone lesions.

In summary, eliglustat blocks RANKL-triggered TRAF3 degradation by preventing autophagic flux. TRAF3 in the cytosol inhibits signaling for the non-canonical NF-KB pathway to induce maturation of OC. Mechanistically, eliglustat significantly reduces the overall amount of GlcCer and LacCer; lipids that are required for autophagy.

**Fig. 5 | Eliglustat inhibits OC formation in a TRAF3-dependent manner.**
**a** RAW264.7 cells were differentiated into OCs with 50 ng/ml M-CSF and 75 ng/ml RANKL. Different doses of eliglustat (0.1, 1, 10, 25, 50 μM) were present throughout the culture period and OCs were identified by TRAP staining on day 5. Scale bar represents 200 μm. Result is representative of four independent experiments. **b**–**e** Number of OCs per well, number of nuclei per TRAP-positive cell, % OC area over total area and Alamar blue viability assay were quantified ($n = 3$ independent experiments). **f** Schematic graph illustrating binding of RANKL to its receptor activates the TRAF6-dependent canonical NF$\kappa$B pathway, leading to proteasome-mediated I$\kappa$B$\alpha$ degradation and nuclear translocation of p65 and p50. In the non-canonical NF$\kappa$B pathway, binding downregulates TRAF3 via autophagy/lysosome-mediated degradation, induces nuclear translocation of p52 and RelB. Panel **f** is adapted from "NF-KB Signaling Pathway" and "Autophagy Process", by BioRender.com (2022). Retrieved from https://app.biorender.com/biorender-templates. **g**–**i** RAW264.7 cells treated with M-CSF, RANKL, and eliglustat for 1 day. TRAF3 protein levels quantified by western blot and mRNA levels quantified by qRT-PCR. Treatment with BafA1 for the last 2 h ($n = 6$ independent samples). **j**–**m** Primary murine BM cells treated with RANKL together with BafA1 or MG132 or eliglustat for 2 h. TRAF6 (**j**, **k**) and I$\kappa$B$\alpha$ (**l**, **m**) levels quantified by western blot ($n = 8$ biologically independent samples). **n** Lethally irradiated 8-week-old female CD45.1$^+$ B6.SJL mice were reconstituted with littermate (WT) or myeloid specific TRAF3 knockout (LysM-Cre + , Traf3 $^{fl/fl}$) BM cells. After 19 days of treatment with eliglustat, bone volume of tibiae was quantified by micro-CT (WT, $n = 4$; WT + Elig, $n = 4$; traf3 KO, $n = 5$; traf3 KO + Elig, $n = 5$ biologically independent animals). The experiment is representative of two independent experiments. Results from **g**, **j**, and **i** are representative of three independent experiments. Data are presented as mean values ± SEM. Exact $p$-values are depicted in the figure. Statistical analysis was performed using one-way ANOVA. Source data are provided as a Source Data file.

This increases TRAF3 volume outside the lysosomal compartment, that ultimately leads to an inhibition of OC formation (Fig. 8i).

## Discussion

As MM bone disease is primarily caused by increased number and overactivation of bone-resorbing OC, compounds that inhibit OC formation or function are of key clinical therapeutic importance. In this study, eliglustat reversed bone loss in C57BL/KaLwRijHsd MM model and C57BL/6J HFD MGUS model as well as demonstrating an additive effect when combined with ZA. By inhibiting OC formation, eliglustat at clinically relevant doses, prevented further bone loss in vivo without affecting OC progenitors, OBs, BMAT, or MM cells.

The combination of eliglustat and ZA prevented osteoclastogenesis and provided extra bone protection in MM mice by using ZA (0.01 mg/kg) at a ten-fold lower concentration than previous reports[54]; thereby allowing further increases in bone volume to be observed. This combination resulted in significantly greater preservation of bone mass than either drug alone, indicating that they act via different mechanisms. This leads to several possible clinical benefits of eliglustat (1) It could be used with ZA in situations where ZA is not completely effective, (2) It could be used as an alternative to ZA if it is not well tolerated, and (3) It could be used in conjunction with ZA for even greater inhibition of bone lesions. This finding is particularly important as a known complication of high dose ZA is ONJ in MM patients[11]. Combination of eliglustat with ZA for optimal bone protection can be achieved with lower amounts of ZA leading to a potential novel clinical strategy with reduced side effects and thus accelerate eliglustat's translational use into the clinic. In addition, following work will be conducted to combine eliglustat with anabolic drugs to further evaluate eliglustat's role in bone disease models.

Based on the potent effect of eliglustat on OC observed in vivo, mechanistic studies were conducted that will underpin future studies in this area. The lipid structure of RAW264.7 cell was more fluid after the administration of eliglustat, which can be achieved by reducing cholesterol or increasing the unsaturated lipids in the plasma membrane[55]. Lipid rafts are a functional membrane unit in charge of signal transduction that are composed of cholesterol and saturated lipids[55]. Ha et al. discovered that lipid raft domains play an essential role in OC differentiation and affect RANK-TRAF6 signaling[56]. Likewise, TRAF3 has been located in the lipid raft microdomain by several groups[57–59]. While this study does not exclude that eliglustat may interfere with other signaling pathways by affecting lipid rafts, eliglustat did not disrupt canonical NF-$\kappa$B signaling as TRAF6 was unaffected. Instead, the lipid structure altered by eliglustat may have an effect on the formation and fusion of intracellular vesicles. Some GSLs including GD3[51], GM2, and GM3 were reported to play an essential role in autophagy[60]. Hence autophagy, required for the degradation of TRAF3, was investigated. The autophagy-dependency of TRAF3 levels has potentially wider clinical applications. In autophagy-deficient A549 cells, elevated TRAF3 levels suppress tumorigenicity[19], suggesting that eliglustat may have antitumor effects in cancers or may be useful to treat diseases with activation of the non-canonical NF-$\kappa$B pathway.

Due to the accumulation of single-membrane autolysosomes rather than double-membrane autophagosomes in RAW264.7 cells treated with eliglustat it is likely that eliglustat prevents autolysosome degradation thereby maintaining TRAF3 levels. Whether this is due to impaired lysosomes remains to be elucidated. For example, lysosomal disruption induced by CQ and other treatments leads to TFEB activation and lysosome biogenesis[61]. In line with our findings, it was reported that the GSL inhibitor D-PDMP acts as an autophagy inhibitor in neuroblastoma cells[62].

CQ and BafA1, similar to eliglustat, both maintain TRAF3 levels and suppress OCs[8,50]. However, CQ has considerable ophthalmic side effects and its substitute hydroxychloroquine, now used more widely in the clinic, had no inhibitory effect on human OC differentiation[8,63]. On the other hand, BafA1 is not an FDA approved drug and its OC inhibition effect was only observed in vitro[50]. Therefore, eliglustat is a more suitable drug than either BafA1 or CQ to treat bone lesions in patients. Our findings pave the way to further investigate if the autophagy inhibitor eliglustat can be used in combination with other compounds to treat diseases beyond bone including refractory MM and melanoma[64,65].

The finding that LacCer and GlcCer are essential for functional autophagy, underscores the vital contribution of lipid membrane composition to autophagy. Furthermore, these GSLs reversed the OC inhibition effect caused by eliglustat. While it was known that some gangliosides induce autophagy[66], this study further identified LacCer and GlcCer (precursors of all gangliosides) to promote autophagy dependent OC differentiation. These GSLs partially rescued autophagy, therefore different concentrations or combinations of lipids may be required to show a complete rescue.

Excitingly, eliglustat inhibited OC formation in patients with MM, and was dependent upon GSLs in human cells. Therefore, there is now tremendous potential for the translational use of eliglustat as a therapeutic in MM bone diseases. Diseases that involve bone loss such as breast cancer and prostate cancer metastasis to bone, postmenopausal osteoporosis and arthritis may all exhibit improvement in bone indices as well. Thus, eliglustat, a clinically approved drug for Gaucher disease, could be repurposed for patients who suffer from various bone loss diseases.

## Methods
### Experimental design
The objective of the study was to investigate if eliglustat increases bone mass and decreases MM bone disease in several preclinical models, and to decipher the underlying mechanism. Using a combination of in vitro assays and in vivo murine models, we revealed the role of eliglustat in the OC differentiation and evaluated the specificity and efficacy of eliglustat. Specifically (i) we used healthy murine models to investigate the cell type that eliglustat affects, (ii) we treated

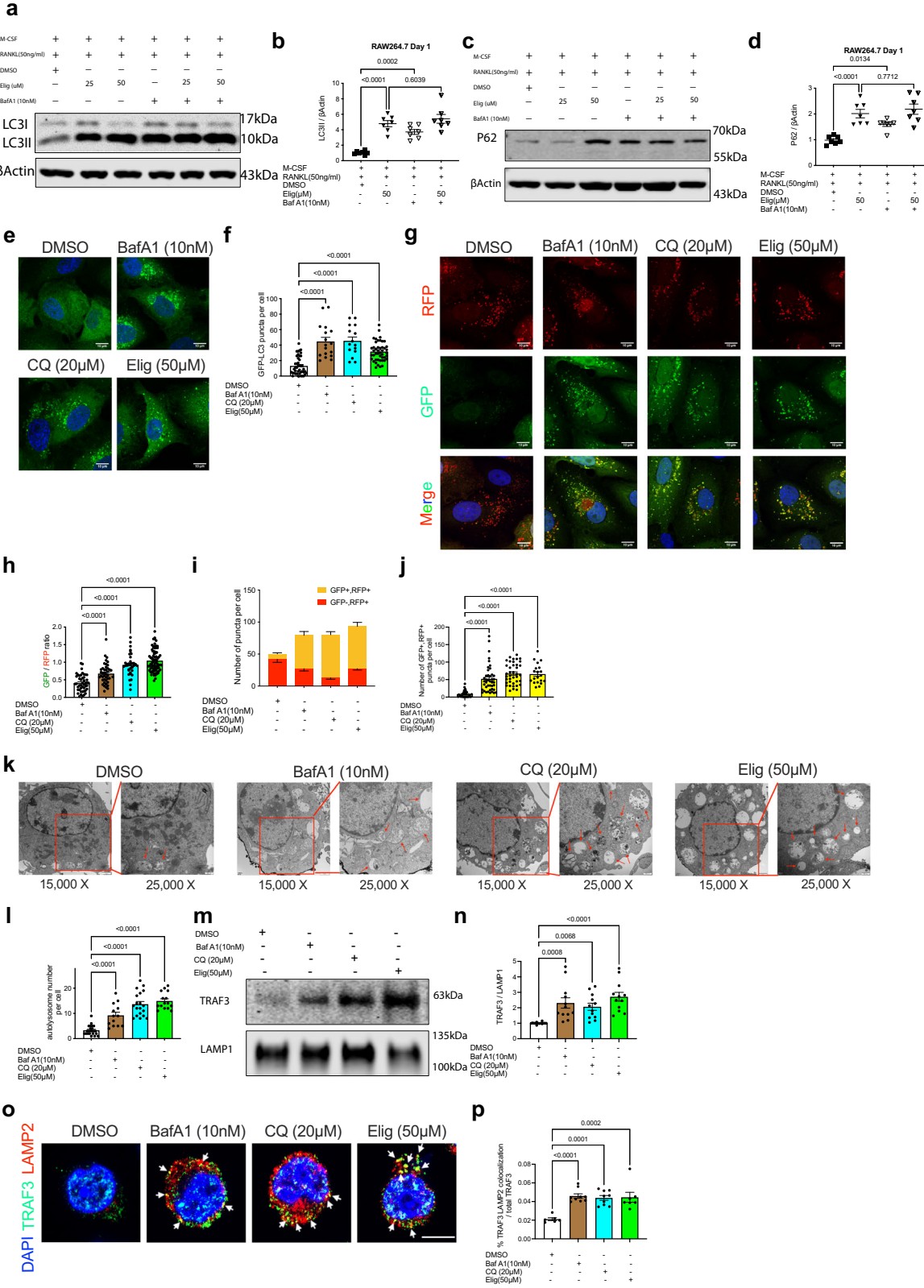

MM and HFD-induced MGUS preclinical models with eliglustat to evaluate its therapeutic potential, (iii) we administrated eliglustat and ZA to MM mice to compare the treatment efficacy, (iv) myeloid specific TRAF3 knockout (LysM-Cre+, Traf3^fl/fl) chimeric mice were used to confirm eliglustat's effect is dependent on TRAF3, and (v) we verified the GlcCer and LacCer findings in OCs derived from MM patients. Sample size was determined by the authors based on power calculations. The exact *n* numbers used in each experiment are indicated in

the figure legends. For in vivo experiments, animals were assigned randomly to the experimental and control groups. Animal allocation, data acquisition and data analysis in vivo or in vitro were performed in a blinded manner.

## Patient samples

The study conforms to the Declaration of Helsinki and Good Clinical Practice guidelines. All human samples collection were reviewed and

**Fig. 6 | Eliglustat is an autophagy inhibitor. a–d** RAW264.7 cells were treated with RANKL and M-CSF with or without Elig for 24 h. 10 nM BafA1 added 2 h before protein harvest. LC3II (**a**, **b**) and p62 (**c**, **d**) levels were quantified by western blot. ($n = 7$ independent samples for DMSO control, Elig, BafA1, and Elig with BafA1). **e**, **f** BafA1, CQ and Elig increased LC3 puncta in U2OS cell transfected with GFP-LC3 plasmid after 2 h treatment ($n = 43$ cells for DMSO control, $n = 17$ cells for BafA1, $n = 15$ cells for CQ, $n = 46$ cells for Elig group). **g–j** Elig increased GFP/RFP ratio (**h**; $n = 46$ cells for DMSO control, $n = 42$ cells for BafA1, $n = 38$ cells for CQ, and $n = 77$ cells for Elig group) and GFP + RFP + puncta number (**j**; $n = 31$ cells for DMSO control, $n = 42$ cells for BafA1, $n = 39$ cells for CQ, and $n = 20$ cells for Elig group) in RFP-GFP-LC3 transfected U2OS cell after 6 h treatment. **k**, **l** Electron microscope images of Elig induced autolysosome (red arrow) formation in RAW264.7 cells after 2 h treatment (25,000×) ($n = 23$ cells for DMSO control, $n = 13$ cells for BafA1, $n = 19$

cells for CQ, and $n = 15$ cells for Elig group) (**l**). **m**, **n** TRAF3 and LAMP1 western blot for lysosome-enriched layer from the control group (DMSO), BafA1 group (10 nM, 2 h treatment), CQ group (20 μM, 12 h treatment), and Elig group (50 μM, 12 h treatment) (**m**); TRAF3 protein level was quantified based on the lysosome marker LAMP1, $n = 12$ independent samples (**n**). **o**, **p** Confocal images of TRAF3 and LAMP2 in Pre-OCs derived from RAW264.7 cells were treated with BafA1, CQ, and Elig for 60 min, Scale bar represents 10 μm, $n \geq 4$ (**o**); the percentage of TRAF3 and LAMP2 co-localization is quantified ($n = 6$ cells for DMSO control, $n = 9$ cells for BafA1, $n = 9$ cells for CQ, and $n = 7$ cells for Elig group) (**p**). Results from **a**, **c**, **e**, **g**, **k**, **m**, and **o** are representative of three independent experiments. Data are presented as mean values ± SEM. Exact $p$-values are depicted in the figure. Statistical analysis was performed using one-way ANOVA. Source data are provided as a Source Data file.

approved by the Ethics Committee of UCL. BM aspirates were donated after informed consent and were consented using the UCL biology of MM consent form. All patients consented to use of their samples for research purposes as part of UCL Biobank study "Biology of Myeloma" (ref 07/Q0502/17). Baseline characteristics of MM patients are shown in Supplementary Table 1. Briefly, BM mononuclear cells were isolated by Ficoll Paque (GE Healthcare) density centrifugation from fresh BM aspirates. No participants were given compensation.

## Cell culture and reagents

5TGM1-GFP, RAW264.7, and U2OS cells were cultured at 37 °C in 5% CO$_2$ with respective complete medium[21,46]. 5TGM1-GFP murine MM cells were provided by B.O. Oyajobi, the University of Texas Health Science Center, San Antonio, and authenticated in C57BL/KaLwRijHsd mouse by serum IgG2bκ ELISA and GFP+ flow cytometry. 5TGM1-GFP cells were routinely maintained in RPMI 1640 medium supplemented with 10% Fetal Bovine Serum, 1% L-glutamine, 1% penicillin/streptomycin, 1% sodium pyruvate, and 1% nonessential amino acids. RAW264.7 cells were obtained from ATCC® (TIB-71™) and authenticated by OC differentiation ability. For in vitro OC assay, murine BM cells, human BM cells or RAW264.7 cells were cultured in Minimum Essential Medium α (α-MEM) with 10% Fetal Bovine Serum, 1% penicillin-streptomycin and 1% L-glutamine differentiated with M-CSF and RANKL as indicated in the figure legends. TRAP experiments were conducted according to published protocol[21]. U2OS reporter cell line was used in the previous publication and authenticated by BafA1 treatment[21,46]; the aliquoted vials were defrosted and cultured in Dulbecco's Modified Eagle Medium supplemented with 10% Fetal Bovine Serum, 1% L-glutamine, and 1% penicillin/streptomycin. All culture media were changed every 3 days. All reagents are listed in Supplementary Table 2.

## Mouse models

Animal experiments were undertaken under UK Home Office Project Licenses 30/3218 and 30/3388 in accordance with the UK Animal (Scientific Procedures) Act 1986. The UK Home Office approved the experiments on animals performed for this study. Customed eliglustat diet was formulated into either a standard rodent diet or a 42% high fat diet, both at 0.075% w/w (150 mg/kg/day for all in vivo experiments) by TestDiet[29]. Sex and age matched C57BL/6J mice and CD45.1+ B6.SJL mice were purchased from Charles River, UK. C57BL/KaLwRijHsd mice were purchased from Harlan, The Netherlands. Myeloid specific TRAF3 knockout (LysM-Cre+, Traf3$^{fl/fl}$) BM cells were used for chimeric mouse generation[8]. All mice were fed and housed under specific pathogen-free level maintenance at 24 °C, 50% humidity and a 12:12 h light/dark cycle in Biological Support Unit at Kennedy Institute of Rheumatology, University of Oxford. Mice were kept in individually ventilated cages, with free access to autoclaved water and irradiated food pellets. Cages were changed once weekly, and the health status of the mice was monitored based on body weight, coat and behavior twice each day. All animals were euthanized in the carbon dioxide euthanasia chamber

with compressed carbon dioxide gas from cylinders inflow to the chamber for 5 min. The death was confirmed by cervical dislocation. Group size was determined based on experimental experience and $n$ numbers are indicated in the figure legends. Littermates from both sexes were randomly assigned, and data acquisition and analysis were performed in a blinded manner.

## Serum and tissue processing

Mouse blood sampling was collected by tail vein bleeding using capillary blood collection tubes (16.440, Microvette® CB 300 Z). These 10ul samples were stored on ice for up to 2 h and centrifuged at full speed for 5 min to extract serum (supernatant). On the day of cull, serum was collected by intracardiac puncture. Serum was stored at −80 °C until further analysis. Mouse spleens were passed through 70 μm cell strainers (542070, Greiner Bio-One) with PBS, 0.1% BSA and 2 mM EDTA to obtain single-cell suspensions. After centrifugation for 3 min at 400 × $g$ in 4 °C, red blood cells were lysed with RBC Lysis Buffer for 5 min at room temperature. RBC-lysed splenocytes were washed once more with PBS, 0.1% BSA, 2 mM EDTA and then used for flow cytometry experiment. Mouse BM cells were collected from the femur. Bones were crushed with a mortar and pestle in PBS-0.1% BSA-2 mM EDTA and filtered through 70 μm cell strainers, then stored on ice ready to be used for flow cytometry without RBC lysis.

## Flow cytometry

Cells were first stained with fixable Zombie Aqua Live/Dead staining and FcR block in PBS for 20 min. Samples were then topped up with PBS-0.1% BSA-2 mM EDTA as a wash. After centrifugation for 3 min at 400 × $g$ in 4 °C, the samples were stained with surface marker antibodies in PBS-0.1% BSA-2 mM EDTA at 4 °C for another 20 min. The cells were washed once with PBS-0.1% BSA-2 mM EDTA. Stained cells were analyzed using four-laser LSR Fortessa X-20 (BD Bioscience). Acquired data was analyzed using FlowJo 10.2.

## Micro-CT analysis

Tibiae were scanned with the PerkinElmer QuantumFX scanner. The region of interest was defined as 100 slices below the growth plate of proximal tibia and was scanned at energy of 90KV; 200uA; field of view 10 for 3 min. Scanned bone data were analyzed and reconstructed by using Analyse-12.0 software.

## TRAP staining and histomorphometry analysis of tibiae

Tibiae fixed in 4% PFA at 4 °C overnight were decalcified in 10% Ethylenediaminetetraacetic acid (EDTA) decalcification solution for 14 days and then paraffin embedded for sectioning with a Leica RM2235 microtome at 5 μm per section to generate histology slides (VWR 631-1553). Sections were processed and stained with TRAP solution and counterstained with 0.2% methyl green solution. The Olympus BX51 microscope or Hamamatsu NanoZoomer S210 equipped with

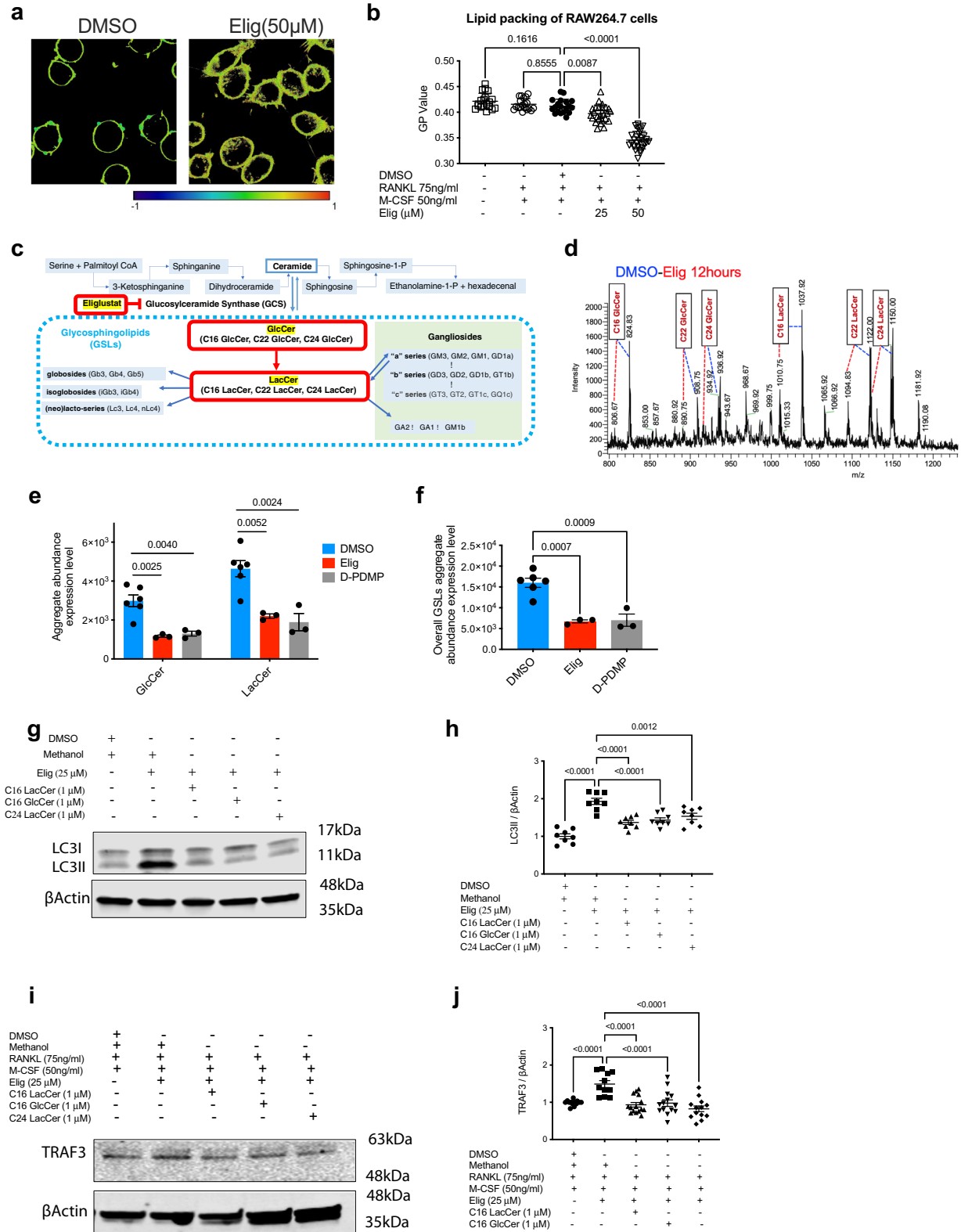

osteomeasure (Osteometrics) software were used to quantify OC parameters. OB parameters and BMAT volume were quantified based on the morphology of OB and adipocyte.

### IgG2bκ ELISA
To evaluate the systemic 5TGM1-GFP MM tumor burden, serum IgG2bκ level was measured by enzyme-linked immunosorbent assay (ELISA) according to the manufacturer's instructions.

### Osmium tetroxide staining for BMAT
Osmium tetroxide stain was used to identify BMAT via micro-CT. Briefly, tibiae were transferred into osmium tetroxide solution for 48 h at room temperature in the fume hood. Then, the tibiae were washed five times with water and embedded in 1% agarose gel. Finally, tibiae are scanned by micro-CT using 90 KV; 200 μA; FOV10 for 3 min and the acquired images used to overlay with previously scanned ones for the bone architecture.

**Fig. 7 | Eliglustat blocks autophagy by inhibiting the GlcCer and LacCer in RAW264.7 cells. a** Membrane fluidity during early OC formation. RAW264.7 cells were treated with RANKL and M-CSF together with eliglustat (25 μM or 50 μM) for 12 h and stained with NR12S dye. Representative images from DMSO group and eliglustat group at 50 μM. Histograms of GP distribution illustrate ordered red end (1) and disordered blue end (−1). Scale bar 10 μm. **b** Quantification for GP value was evaluated (*n* = 19 control cells, *n* = 19 RANKL + M-CSF, *n* = 19 DMSO + RANKL + M-CSF, *n* = 25 RANKL + M-CSF + 25 μM Elig, *n* = 28 RANKL + M-CSF + 50 μM Elig group). **c** Schematic of lipid, GSL and ganglioside metabolism. Eliglustat blocks GlcCer formation from ceramide, subsequently blocking formation of LacCer, Gangliosides and other GSLs (including globosides, isoglobosides, and [neo]lacto-series). **d** LTQ-ESI-MS glycosphingolipidomics profile of RAW264.7 cell from DMSO group and eliglustat group (50 μM, 12 h) with 1:1 sample mix for change of specific GSL composition (blue dash line indicates DMSO, red dash line indicates eliglustat) (*m/z* means *mass charge ratio*). **e, f** GlcCer (cluster of molecular ion peaks between *m/z* 807 and 935; including C16, C22, C24) and LacCer (cluster of molecular ion peaks between *m/z* 1011 and 1150; including C16, C22, C24) aggregate abundance in DMSO, eliglustat and D-PDMP groups (**e**). Overall GSLs including GlcCer, LacCer and Lc3 (**f**); *n* = 3–6 independent samples per group. **g, h** RAW264.7 cells were treated with 25 μM eliglustat and 1 μM C16 LacCer, 1 μM C16 GlcCer, or 1 μM C24 LacCer for 24 h (**g**); LC3II levels were quantified by western blot, *n* = 8 independent samples per group (**h**). **i, j** RAW264.7 cells were treated with RANKL and M-CSF in addition to eliglustat and GSLs for 24 h (**i**). TRAF3 levels were quantified by western blot; RANKL + M-CSF, *n* = 14; RANKL + M-CSF + 25 μM Elig, *n* = 11; RANKL + M-CSF + 25 μM Elig + 1 μM C16 LacCer, *n* = 15; RANKL + M-CSF + 25 μM Elig + 1 μM C16 GlcCer, *n* = 14; RANKL + M-CSF + 25 μM Elig + 1 μM C24 LacCer, *n* = 13 independent samples (**j**). Results from **a**, **d**, **g**, and **i** are representative of three independent experiments. Data are presented as mean values ± SEM. Exact *p*-values are depicted in the figure. Statistical analysis was performed using one-way ANOVA. Source data are provided as a Source Data file.

## Western blot

Cells were lysed using NP-40 lysis buffer containing proteinase inhibitors (Sigma) on ice. The protein concentration was quantified by BCA Assay (23227, Thermo Fisher). Reducing Laemmli Sample Buffer was then added to the lysate and heated at 95 °C for 10 min. 15–30 μg proteins were loaded for SDS-PAGE analysis. NuPAGE Novex 4–12% Bis-Tris gradient gel with MES running buffer (Thermo Fisher) was used. To improve separation of LC3-I and LC3-II, 15% Tris-HCl gel and SDS running buffer was used. Proteins were transferred to a PVDF membrane (IPFL00010, Merck Millipore) and blocked with 5% skimmed milk-PBST. Membranes were incubated with primary antibodies dissolved in 1% milk overnight and secondary antibodies dissolved in 1% milk with 0.01% SDS for 1 h and then imaged using the Odyssey CLx Imaging System. The signal from IRDye 800 and IRDye 680 were acquired simultaneously but with different intensity based on the fluorescent exposure of the bands. Data were analyzed using Image Studio Lite. Uncropped and unprocessed scans of the blots are shown as a supplementary Fig. 7 in the Supplementary Information.

## Quantitative PCR

Polymerase chain reaction (PCR) was used to detect the change in gene expression in cells. RNA was extracted using the RNeasy Plus Mini Kit (74134, Qiagen). The concentration of RNA was measured using NanoDrop1000 followed by reverse transcribed the RNA to cDNA using High capacity RNA-to-cDNA Kit (Thermo Fisher, 4387406). Taqman probes (Thermo Fisher) and TaqMan Gene Expression Master Mix (Thermo Fisher, 4369016) were used for the qPCR reactions according to the manufacturer's instructions. The ViiA 7 Real-Time PCR System was used to conduct the quantitative PCR. The quantification of target mRNAs expression was based on gapdh. ΔΔCt method was used in all of the experiments.

## Chimera blood lineages analysis

BM chimera was conducted on 11 Gy lethally irradiated 8-week-old female CD45.1+ B6.SJL mice using BM cells from CD45.2 OC-specific deletion of *traf3* female mice or their littermates. Reconstitution efficiency was confirmed by evaluating CD45.2 cells in the CD45.1 mice 6-week post-irradiation[67]. Fifty microliters of blood from tail vein bleeding was collected and treated with RBC Lysis Buffer for 5 min. After centrifugation, the supernatant was removed and cells were stained with antibodies for flow cytometry (Supplementary Table 3).

## Immunofluorescence staining

For TRAF3 and LAMP2 staining, cells were fixed with 4% paraformaldehyde for 10 min and permeabilized with 0.2% Triton X-100 in PBS for another 10 min in room temperature. Cells were blocked with 2% BSA-PBS for 1 h and incubated with anti-LAMP2 antibody in 2% BSA-PBS for overnight at 4 °C. After washing with PBS for three times, cells were incubated with TRAF3 antibody and AF647 secondary antibody in 2% BSA-PBS for overnight. After washing with PBS and stained with 2 ug/ml DAPI for 10 min, coverslips were mounted in SlowFade™ Diamond Antifade Mountant (ThermoFisher, S36963). Optical sections were acquired using a Zeiss LSM 980 Airyscan confocal system. Co-localization analyses were performed using *arivis Vision4D X64* software. In short, the co-localization of TRAF3 and LAMP2 was identified and counted when the TRAF3 region showed a ≥50% area overlap with LAMP2. To show the TRAF3-LAMP2 co-localization precisely, 1 slice out of 11 slices stack (each image with 2 μm distance) from each of the cell is shown in the figure. For LAMP2 staining in the supplemental figures, cells were fixed using 4% paraformaldehyde and incubated with 0.1% saponin and 10% FBS for 60 min. Cells were then incubated in primary antibody with 10% FBS for overnight at 4 °C and secondary antibody for 1 h then stained with DAPI for 5 min. Coverslips were added with mounting medium prior to examination under Olympus Fluoview FV3000 or Leica TCS SP8. LAMP2 area analysis was performed using MATLAB software.

## Transmission electron microscopy

$1 \times 10^8$ RAW264.7 cells were fixed with 0.1 M sodium cacodylate buffer solution (pH = 7.4) containing 2.5% glutaraldehyde for 1 h at 4 °C. Cells were then scraped and centrifuged. Samples were submitted to the Core Facility at Shanghai Jiao Tong University for ultrastructural analyses (HITACHI 7650).

## Lysosome isolation

The lysosome isolation procedures were conducted according to manufacturer's protocol. In short, RAW264.7 cells were lysed in a Dounce homogenizer (named as whole-cytosolic lysate) and loaded onto 19% Optiprep solution for ultracentrifuge at 150,000 × *g* for 4 h. The lysosome layer was then collected for western blot.

## Spectral imaging

RAW264.7 cells were seeded on glass bottom dishes. After treatment, cells were incubated with 1 μM NR12S dye for 5 min. Spectral imaging was performed by Zeiss LSM 780 microscope, WIMM, University of Oxford. Laser light at 488 nm was selected for NR12S excitation and the λ-detection range was set between 490 and 691 nm. Images were analyzed with the custom GP plugin of Fiji software as described[68].

## GSLs extraction and methylation

$5 \times 10^7$ RAW264.7 cells were sonicated in chloroform/methanol 1:1 (*v:v*) for three times and another three times with isopropanol/hexane/water 55:25:20 (*v:v:v*) in glass tubes. After centrifugation, the supernatant was collected and dried with centrifugal vacuum. Anion-exchange chromatography of DEAE Sephadex A-25 (Sigma) was used

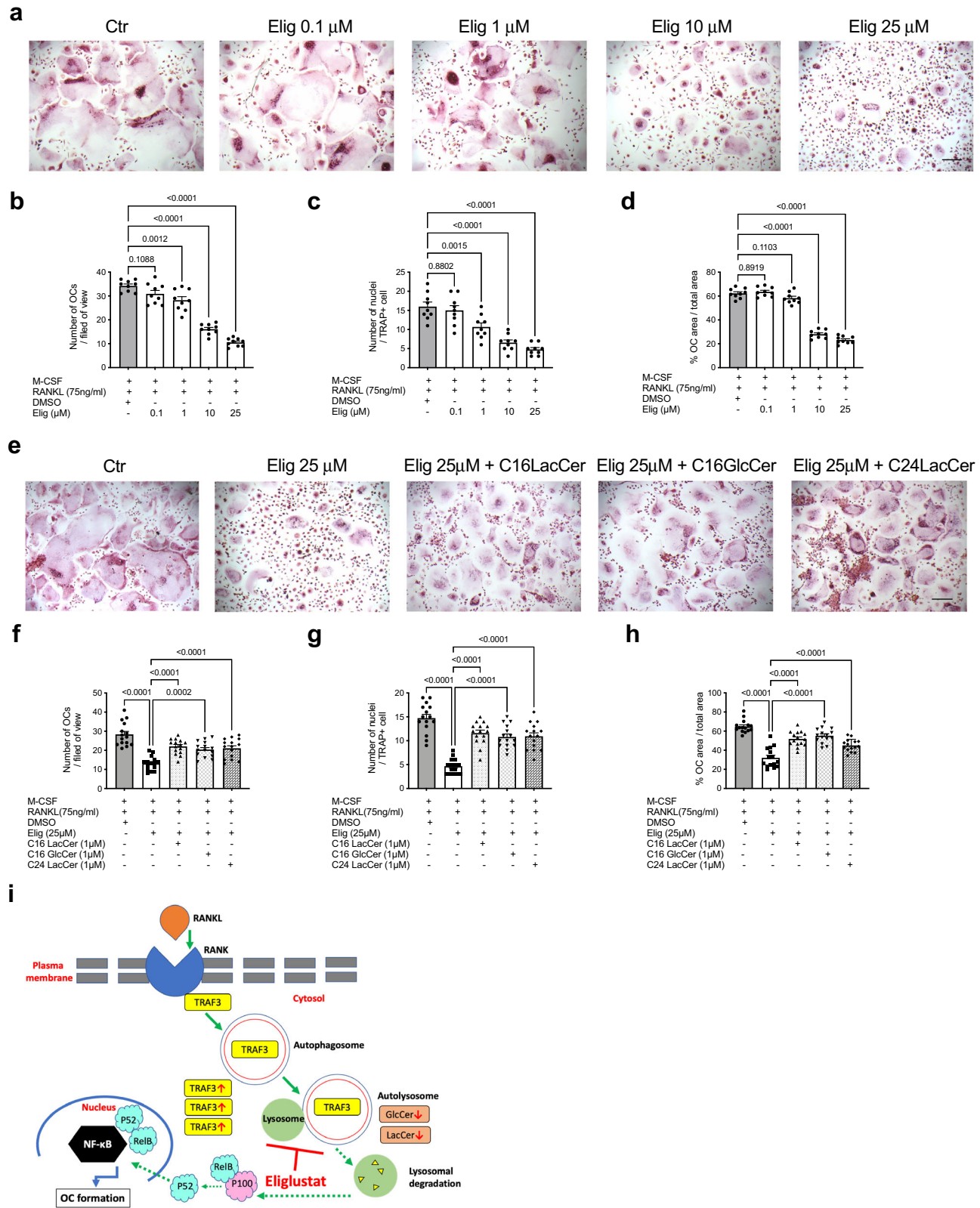

to separate the total dried crude lipids, the neutral lipid fraction was enriched by chloroform/methanol/water 30:60:10 (*v:v:v*). Neutral lipids were then dissolved with sodium hydroxide (Iodomethane-d₃ was added to DMSO group). After quenched with 2 ml water, the permethylated products were extracted by adding 2 ml of dichloromethane and washed with water. The dichloromethane phase was then collected and dried under vacuum[69,70].

## Analysis of permethylated GSLs by LTQ-ESI-MS

The samples were dissolved in methanol-acetonitrile 20:80 (v:v). In order to show GSL profile of the Iodomethane-d3 labeled DMSO-control samples and the treated samples on the same graph, samples were mixed in equal volume for mass spectrometry. We noticed this brings the isotope peak of Iodomethane-d3 labeled C22LacCer main peak interfered the main peak of C24 LacCer due to. Hence, we did not

**Fig. 8 | Eliglustat inhibits OC formation from MM patients while exogenous GSLs reverse eliglustat inhibition. a** BM mononuclear cells from MM patients were seeded at $1.5 \times 10^5$ cells/well into 48-well-plate and differentiated to OCs with 50 ng/ml human M-CSF and 75 ng/ml human RANKL in αMEM media. BM mononuclear cells were treated with increasing doses of eliglustat (0.1, 1, 10, 25 μM) and OCs were identified by TRAP staining on day 11. **b**–**d** Number of OCs per filed of view, number of nuclei per TRAP + cell and % OC area over total area were quantified. $n = 9$ patients. **e** Eliglustat was applied to the BM mononuclear cells from MM patients with or without C16 LacCer, C16 GlcCer, and C24 LacCer during in vitro OC formation process and mature OCs were recognized by TRAP staining. **f**–**h** Number of OCs per filed of view, number of nuclei per TRAP + cell and % OC area over total

area were quantified. Scale bar represents 200 μm. $n = 9$ patients, each with 5–7 replicates. **i** Schematic graph illustrating the overall proposed mechanism. Binding of RANKL to RANK receptor induces the degradation of TRAF3 via autophagy, which allows NF-kB signaling to occur; this is required for OC formation. Eliglustat reduces the amount of GlcCer and LacCer, preventing autophagic flux, thereby reducing TRAF3 degradation and OC formation. Results from **a** and **e** are representative of 3 independent experiments with 9 patient samples. Data are presented as mean values ± SEM. Exact $p$-values are depicted in the figure. Statistical analysis was performed using one-way ANOVA. Source data are provided as a Source Data file.

include C24 LacCer for statistical quantification. Analyses were performed with High Performance Liquid Chromatography (HPLC) and a linear ion-trap mass spectrometer (LTQ-XL, Thermo Finnigan, USA). HILIC column was used with flow rate at 50 uL/min, and the effluent was infused into the ion-trap mass spectrometer. The MS parameters were as follows: ion spray voltage 3.5 kV, sheath gas flow rate 2 arbitrary units, capillary voltage 35 V, capillary temperature 300 °C, tube lens 110 V, injection time 100 ms, activation time 30 ms, and isolation width $m/z$ 1.5. Full Original Lipidomics Data is available in Supplementary Data 1.

### Statistical analyses

All data shown are represented as mean ± standard error of the mean (SEM). Two-tailed $t$-test was applied when comparing two experimental groups. For comparisons between two normally distributed datasets with equal variances, unpaired two-tailed Student's $t$-test was applied. For the experiments containing ≥3 experimental groups, one-way analysis of variance (ANOVA) and multiple comparisons were used with Tukey's correction. Paired or unpaired one-way ANOVA was used for multiple comparisons of normally distributed datasets with one variable. To evaluate the statistical significance of the hypothesis being tested, $p$-value was applied. Statistical analyses were performed using GraphPad Prism (San Diego, CA).

### Reporting summary

Further information on research design is available in the Nature Portfolio Reporting Summary linked to this article.

## Data availability

The data supporting the findings of this study are available within the paper and its Supplementary Information files and Source Data file. The original lipidomics data generated in this study are provided in the Supplementary Information file. Source data are provided with this paper.

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

## Acknowledgements
We thank the patients who donated clinical samples used in this research. Zhenqiang Yao and Akram Ayoub from Brendan Boyce's lab in University of Rochester Medical Center who provided scientific advice and prepared the TRAF3 KO mouse BM cells—it is greatly appreciated. We thank Xiaoxia Liu and members from Qing Zhong's lab who helped with experiments. Confocal microscopy help from Christoffer Lagerholm, Senior Advanced Microscopy Manager in the Kennedy Institute of Rheumatology, is appreciated. Bin Xie, Pengjun Xi, and Jianqing Zheng from Kennedy Institute of Rheumatology, Zhu Liang from Target Discovery Institute, Jiahao Jiang and Jianwei Cui from Wellcome Centre for Human Genetics, University of Oxford; Jin Wang and Duohui Jing from Shanghai Ruijin Hospital; Ge Zhang from Hong Kong Baptist University all provided constructive comments. This work was supported by University of Oxford Medical and Life Sciences Translational Fund MC_PC_17174 and MC_PC_18059 from Wellcome ISSF fund and the MRC confidence in concept to A.K.S. and H. Leng; Wellcome Trust Fund 220784/Z/20/Z to A.K.S.; Versus Arthritis grant 20372 to N.J.H. and A.E.; Genzyme grant GZ-2015-11433 to N.J.H.; NSFC grant 91754205, 91957204, and 31771523 to Q.Z.; China Scholarship Council to H. Leng; China Scholarship Council-Nuffield Department of Medicine Scholarship and the Oxford-Elysium Prize Fellowship to H.Z.; Blood Cancer UK 17012 and 15026 to C.M.E. and E.V.M.; Knut and Alice Wallenberg Foundation, Swedish Research Council (2020-02682) and Wellcome Trust ISSF to E.S.; European Molecular Biology Organization Postdoctoral Fellowship (EMBO) ALTF115-2019 to A.V.L.V.

## Author contributions
H. Leng, H.Z., A.K.S., and N.J.H. conceptualized and designed the study. H. Leng, H.Z., L.L., S.Z., Y.W., S.J.C., D.G.-F., E.S., and A.V.L.V. performed and analyzed experiments, methodology, and investigation. H. Leng, H.Z., L.L., S.Z., Y.W., S.J.C., D.G.-F., H. Lou, A.E., E.V.M., E.S., Y.-H.L., Y.L., A.V.L.V., M.T., J.M., and K.Y. discussed analyses. H. Leng, H.Z., Q.Z., C.M.E., A.K.S., and N.J.H. supervised the project and interpreted the experimental data. H. Leng, A.K.S., and N.J.H. wrote the original draft. H. Leng, H.Z., C.M.E., A.K.S., and N.J.H. reviewed and edited the manuscript.

## Competing interests
The authors declare no competing interests.

## Additional information

¹Kennedy Institute of Rheumatology, University of Oxford, Roosevelt Drive, Oxford OX3 7FY, UK. ²Key Laboratory of Cell Differentiation and Apoptosis of Chinese Ministry of Education, Department of Pathophysiology, Shanghai Jiao Tong University School of Medicine, Shanghai, P.R. China. ³Computational Biology Department, Carnegie Mellon University, Pittsburgh, PA 15217, USA. ⁴Institutes of Biology and Medical Sciences, Soochow University, Suzhou, P.R. China. ⁵Department of Hematology, UCL Cancer Institute, University College London, London, UK. ⁶Ludwig Institute for Cancer Research, Nuffield Department of Medicine, University of Oxford, Oxford OX3 7DQ, UK. ⁷Norwich Medical School, University of East Anglia, James Watson Road, Norwich NR4 7UQ, UK. ⁸Nuffield Department of Surgical Sciences, Botnar Research Centre, University of Oxford, Old Road, Oxford OX3 7LD, UK. ⁹Science for Life Laboratory, Department of Women's and Children's Health, Karolinska Institute, Solna, Sweden. ¹⁰MRC Weatherall Institute of Molecular Medicine, MRC Human Immunology Unit, Oxford OX3 9DS, UK. ¹¹Human Phenome Institute, Fudan University, 825 Zhangheng Road, Shanghai, P.R. China. ¹²Shanghai Institute of Hematology, State Key Laboratory of Medical Genomics, National Research Center for Translational Medicine at Shanghai, RuiJin Hospital Affiliated to Shanghai Jiao Tong University School of Medicine, Shanghai, P.R. China. ¹³Nuffield Department of Orthopaedics, Rheumatology, and Musculoskeletal Sciences, Botnar Research Centre, University of Oxford, Old Road, Oxford OX3 7LD, UK. ¹⁴These authors contributed equally: Anna Katharina Simon, Nicole J. Horwood. ✉e-mail: katja.simon@imm.ox.ac.uk; n.horwood@uea.ac.uk

