## [Peer Review File · Nature Communications]

Modulating glycosphingolipid metabolism and autophagy improves outcomes in pre-clinical models of myeloma bone diseaseREVIEWER COMMENTS

Reviewer #1 (Remarks to the Author):

The manuscript entitled “Improving bone health via modulation of glycosphingolipid metabolism and autophagy” by Leng et al. demonstrated that eliglustat treatment inhibits osteoclast differentiation and excessive bone resorption in preclinical models of multiple myeloma. The proposed that the mechanism of action of Eliglustat is dependent on the suppression of TRAF3 degradation via lysosome/autophagy pathway. In general, the paper is easy to read and well performed. Eliglustat is an FDA approved glucosylceramide synthase inhibitor, thus the clinical relevance of these finding is potential very high. The in vivo data showing rescue of bone loss in eliglustat-treated models of MM are convincing.

However, in my view the data supporting the mechanism of action of Eliglustat should be strengthened by additional experiments.

IN particular:

- 1) There is a clear effect of Eliglustat on lysosome function. However, the effects of eliglustat on lysosomal trafficking and accumulation of TRAF3 in RAW cells has not been investigated in sufficient details. Live imaging/ time course and confocal microscopy analysis of TRAF3 subcellular localization in different conditions should be performed to visualize Eliglustat roles on TRAF3 in stimulated vs unstimulated osteoclasts.
- 2) The use of additional pharmacological inhibitors autophagy (e.g. inhibitors of AV biogenesis) is needed to support the proposed mechanisms.
- 3) It is important to characterize the defects of autophagy induced by Eliglustat in osteoclasts. It appears to me that Eliglustat is inhibiting lysosomal function, that is essential for osteoclast activity per se, since it has fundamental roles during bone resorption. Is TRAF3 accumulating in lysosomes of Eliglustat treated cells? If this is the case, why is it still functioning?

In summary I recommend the authors to improve the characterization of Eliglustat roles on TRAF3 degradation and on lysosomal function.

Additional (minor) points:

Figure 6F: In addition to LC3 the authors should perform additional experiments to support the rescue by LacCer and GlcCer molecules

Figure 6G: the blot is not convincing. There is no clear evidence of TRAF3 variation between different treatments.

Reviewer #2 (Remarks to the Author):

This is an interesting manuscript that explores the ability of eliglustat to prevent osteoclast formation and bone loss in both murine models of myeloma and samples isolated from patients. The data is interesting and points to a new approach to targeting osteoclasts that could work together with conventional approaches. The manuscript is therefore both interesting from a mechanistic angle but also from a translational perspective. The data is compelling although as presented there are areas that could be developed further to better articulate the rationale.

The story is really about myeloma bone disease with potential implications for broader bone health so maybe title should reflect this. Alternatively, much more should be made of the supplementary data looking at eliglustat in normal mice in the absence of tumor

The abstract appears to reverse the rationale used in the narrative that runs through the paper so in some ways it appears inconsistent confusing. Reordering the abstract to mirror the flow of the manuscript may make it easier to transition from abstract to the body of the manuscript

Part of the rationale used for this study is the fact that current agents can be associated with side-effects – although this may reflect the cumulative dose. In the introduction there is reference to bisphosphonates. Acknowledging that RANKL targeted reagents are also considered for patients with myeloma would add balance without distracting from the rationale behind the study.

In the Introduction \ a clearer articulation of autophagy and the role of the various genes would help those not in the field.

The in vivo studies are interesting and the outcomes are clear. It was unclear why the control studies in the absence of tumor were performed separately and included as supplementary data – particularly for the data in figures 1 and 2 – yet were included in the figure 3 – the latter appears more compelling.

Numbers of mice are small in some studies (n=4). It would help to understand whether these studies were repeated. It is also interesting that BV/TV is high in the control MM group in figure 2D probably several times higher than that seen in Figure 1. Is there an explanation for this.?

The section on the high bone mass diet was hard to follow. The data is interesting but articulation in the results section was hard to follow.

The data on traf3 $-/-$ is modest with relatively few animals and limited analysis. It would be good to see a more detailed analysis in the same way as shown in figures 1-3.

It was unclear whether the patient data is from a single patient. I seem to have missed the supplementary table.

The authors argue that this agent could be used more broadly. It would be good to understand whether there are potential significant off-target effects anticipated – presumably they would be limited if this is used in other clinical settings.

Reviewer #3 (Remarks to the Author):

In this paper the authors reported the involvement of glycosphingolipid in autophagy during osteogenesis. The presented results are potentially interesting and deals with a valuable topic. However, there are some ambiguous parts that need to be organized and make the different parts of the article fluid together and, in addition, there are some English errors that make some parts unclear.

In particular these are some comments and concerns need to be addressed:

1) lines 40-41 in abstract are not so clear and appear not linked with the first sentence. Generally the abstract is not appealing.

2) line 75, the information about TRAF3 degradation is very general, as the results of this project are focused on autophagy and TRAF3 degradation, it is important to explain it in more detail in both introduction and also discussion sections.

3) line 100, the paragraph title is very general, as the authors have explained lots of results in this section, it is better to write a more detailed title too.

4) Why did the authors not evaluate Eliglustat on normal control animals to understand the effects of this substance in normal conditions? They have evaluated this drug only on MM group. This must be included.

5) Why have the female mice been selected for this study while the hormonal cycles can interfere with the experiment and in particular with bone structure? I suggest to evaluate the differences with male mice, if any.

6) What is the rational explanation for studying the effects of Eliglustat on HFD bone markers? It should be explained.

7) What is the role of MG132 on the OC differentiation pathway? In the figure 4 there is information about the role of MG132 on I κ B α degradation and also OC precursors have been treated with this substance, but in the text the authors did not indicate its role and why they used it.

8) The differentiation of Raw cells into OC is a long process that needs at least 1 week. Why did they treat and evaluate cells only for 1 day (line 264)?

9) line 357, it is necessary to indicate "mononuclear cells" isolated from BM instead of "BM" alone that is not clear.

Dear Editor,

We are very grateful for the helpful and constructive reviewer's comments. The authors hope that our response to the suggested amendments will meet your requirement for publication.

REVIEWER COMMENTS

Reviewer #1 (Remarks to the Author)

The manuscript entitled "Improving bone health via modulation of glycosphingolipid metabolism and autophagy" by Leng *et al.* demonstrated that eliglustat treatment inhibits osteoclast differentiation and excessive bone resorption in preclinical models of multiple myeloma. The proposed that the mechanism of action of Eliglustat is dependent on the suppression of TRAF3 degradation via lysosome/autophagy pathway. In general, the paper is easy to read and well performed. Eliglustat is an FDA approved glucosylceramide synthase inhibitor thus the clinical relevance of these finding is potential very high. The *in vivo* data showing rescue of bone loss in eliglustat-treated models of MM are convincing. However, in my view the data supporting the mechanism of action of Eliglustat should be strengthened by additional experiments.

In particular:

1) There is a clear effect of Eliglustat on lysosome function. However, the effects of eliglustat on lysosomal trafficking and accumulation of TRAF3 in RAW cells has not been investigated in sufficient details. Live imaging/ time course and confocal microscopy analysis of TRAF3 subcellular localization in different conditions should be performed to visualize Eliglustat roles on TRAF3 in stimulated vs unstimulated osteoclasts.

To image the location of TRAF3 in RAW264.7 cells, a TRAF3 monoclonal antibody was used to see the overall TRAF3 distribution in the cell. As kindly suggested by the reviewer, we conducted a time course (0, 15, 30, 60 and 120 minutes) using confocal microscopy analysis of the subcellular localization of TRAF3 together with LAMP1 in the presence and absence of the autophagy inhibitors BafA1, CQ or eliglustat (Figure 6E-H). Cells treated with RANKL and M-CSF in the absence of an autophagy inhibitor were used as a control.

This experiment shows that TRAF3 colocalized with LAMP1 indicating that TRAF3 targets the lysosome; this is in agreement with published results in Newman *et al.* *Nat Comms* 2017⁸. Furthermore, non-colocalised TRAF3 in the cytosol increased over time indicating that there is more TRAF3 being retained rather than being degraded.

The previous Figure 6E (LAMP2 alone) has been moved to become Figure S6B whilst the colocalization of TRAF3 with LAMP1 and the quantification of TRAF3 and LAMP1 volume have been included in the main figures as the new Figure 6E-H.

In addition, we also prepared a schematic graph to illustrate the mechanistic findings. In summary, eliglustat blocks RANKL-triggered TRAF3 degradation by preventing autophagic flux. TRAF3 in the cytosol inhibits signalling for the non-canonical NF- κ B pathway to induce

maturation of OC. Mechanistically, eliglustat significantly reduces the overall amount of GlcCer and LacCer, which are lipids that are required for autophagy. This increases TRAF3 volume outside the lysosomal compartment, which ultimately leads to the inhibition of OC formation (Figure 8E).

Figure 6E-H.

Figure 8E.

2) The use of additional pharmacological inhibitors of autophagy (e.g. inhibitors of AV biogenesis) is needed to support the proposed mechanisms.

Using a series of different autophagy inhibitors (see schematic adapted from Wang *et al.*, 2016¹), we now show that all were able to dose dependently inhibit osteoclast formation (Figure S6C, D). 3MA (a VPS34 inhibitor and class I PI3K inhibitor) and SAR405 (a VPS34 inhibitor), both of which inhibit nucleation process in autophagy, were used to see if other autophagy inhibitors upstream prevent OC formation^{1,2}. Whilst BafA1 and CQ, both lysosomal inhibitors that mechanistically work similarly to eliglustat and disrupt the completion process of autophagy^{1,2}, likewise inhibited OCs. This is consistent with previous findings that 3MA³, CQ⁴ and BafA1⁵ attenuate OC formation.

Schematic adapted from Wang *et al.*, 2016¹.

Figure 6C-D.

3) It is important to characterize the defects of autophagy induced by Eliglustat in osteoclasts. It appears to me that Eliglustat is inhibiting lysosomal function, that is essential for osteoclast activity per se, since it has fundamental roles during bone resorption. Is TRAF3 accumulating in lysosomes of Eliglustat treated cells? If this is the case, why is it still functioning?

Firstly, TRAF3 localisation was determined using a cell fractionation method to separate nucleus fraction, total cytosolic fraction, C1 fraction including soluble cytosolic fraction without organelles (endoplasmic reticulum, golgi apparatus, mitochondria) and organelle fraction (mitochondria). In the eliglustat treated group, both cytosolic and C1 TRAF3 increased (Figure S6A) however a definitive conclusion could not be reached due to lysosomal contamination of the cytosolic fraction.

Subsequently, time course confocal experiments (Figure 6E-H) as described for point 1 of the rebuttal showed that TRAF3 was indeed associated with the lysosomes but that it also continued to accumulate in the cytoplasm. As such it is hypothesized that TRAF3 is still able to inhibit osteoclastogenesis due to an excess in the cytoplasm.

Supplemental Figure 6

Figure S6A.

Figure 6E-H.

In summary I recommend the authors to improve the characterization of Eliglustat roles on TRAF3 degradation and on lysosomal function.

Additional (minor) points:

Figure 6F: In addition to LC3 the authors should perform additional experiments to support the rescue by LacCer and GlcCer molecules

In Figure 8, we demonstrate that LacCer and GlcCer are able to rescue human OC formation using bone marrow sample precursors from myeloma patients (Figure 8A-D).

Figure 6G: the blot is not convincing. There is no clear evidence of TRAF3 variation between different treatments.

Below are the rest of the representative original blots (quantification shown in blue; used to generate the TRAF3 over actin ratio). Whilst the rescue effect of LacCer and GlcCer in original membranes is subtle, the quantification has been rigorously undertaken to ascertain the validity of our conclusion. Following the reviewer's comment, we have changed the representative blot and now use the second part of Membrane 7 for display in the current main Figure 7G.

Other Membrane 1

Other Membrane 2

Other Membrane 3

Other Membrane 4

Other Membrane 5

Other Membrane 6

Other Membrane 7

Previous 7G Membrane 8

Figure 7G-RAW264.7 with RANKL and M-CSF

Figure 7G-RAW264.7 with RANKL and M-CSF

Reviewer #2 (Remarks to the Author):

This is an interesting manuscript that explores the ability of eliglustat to prevent osteoclast formation and bone loss in both murine models of myeloma and samples isolated from patients. The data is interesting and points to a new approach to targeting osteoclasts that could work together with conventional approaches. The manuscript is therefore both interesting from a mechanistic angle but also from a translational perspective. The data is compelling although as presented there are areas that could be developed further to better articulate the rationale.

1. The story is really about myeloma bone disease with potential implications for broader bone health so maybe title should reflect this. Alternatively, much more should be made of the supplementary data looking at eliglustat in normal mice in the absence of tumour.

The title has been changed to reflect the emphasis on myeloma bone disease. The new title is 'Improving myeloma bone disease via modulation of glycosphingolipid metabolism and autophagy.'

Supplemental Figure 1 in the original manuscript has been moved into the main body of the article as Figure 1 to show the effects of eliglustat in normal mice in the absence of tumour. New text is shown in red throughout the manuscript.

2. The abstract appears to reverse the rationale used in the narrative that runs through the paper so in some ways it appears inconsistent/confusing. Reordering the abstract to mirror the flow of the manuscript may make it easier to transition from abstract to the body of the manuscript.

Many thanks for this direction, the abstract has been reordered in accordance with the order of data presentation in the main body of the text (lines 42-54).

3. Part of the rationale used for this study is the fact that current agents can be associated with side-effects – although this may reflect the cumulative dose. In the introduction there is reference to bisphosphonates. Acknowledging that RANKL targeted reagents are also considered for patients with myeloma would add balance without distracting from the rationale behind the study.

Additional text has been added to the introduction (lines 66-69) to include discussion of denosumab.

4. In the introduction, a clearer articulation of autophagy and the role of the various genes would help those not in the field.

Additional description has been added to aid the reader with the process of autophagy (lines 75-85).

5. The *in vivo* studies are interesting and the outcomes are clear. It was unclear why the control studies in the absence of tumour were performed separately and included as

supplementary data – particularly for the data in figures 1 and 2 – yet were included in the figure 3 – the latter appears more compelling.

For all in vivo experiments, ways to ensure responsible experimental animal use was undertaken to comply with 3R principles. Hence animal numbers were kept to the minimum required for statistically significant results as well as keeping control group repetition to those necessary for data interpretation.

Supplemental Figure 1 in the original manuscript has been moved into the main text as Figure 1 to show the effects of eliglustat in normal mice in the absence of tumour. Figure 2 shows mice without tumour (control), those with MM and then the effects of eliglustat on MM. As Figure 1 shows the effect of eliglustat without tumour, it was not necessary to include another experimental group in Figure 2.

In Figure 3, there is a change of mouse model to the C57BL/6J mice. These mice are not susceptible to 5TGM1 cell engraftment unless fed on the high fat diet (HFD), irradiated, or in old age. As such, the mice injected with 5TGM1 (MM) cells are the negative controls for this experiment, the HFD + MM mice are the positive control and the HFD + MM + Elig are the experimental group.

In Figure 4, there were 5 experimental groups (control, MM, MM + Elig, MM + ZA, and MM + Elig + ZA) as the mice used were male whereas all former experiments were done in female mice. As such it was important to include a 'no tumour' control group as bone indices vary between males and females. Experiments in male mice that investigate effects of eliglustat in the absence of tumour (to complement Figure 1) and in MM-bearing mice (to complement Figure 2) have been included as supplemental Figures S1A-B and S2A-B, respectively.

6. Numbers of mice are small in some studies (n=4). It would help to understand whether these studies were repeated. It is also interesting that BV/TV is high in the control MM group in figure 2D probably several times higher than that seen in Figure 1. Is there an explanation for this?

As described in the previous comment, where possible, animal numbers were kept to the minimum required for statistically significant results. In Figure 3D for the HFD model in C57BL/6J mice, n=4 was used in the negative control group (where no MM engraftment occurs) to compare to the positive control HFD group (n=10) to ensure that the diet was effective. This is in agreement with previous findings from our laboratories⁶.

For Figure 5H, n=4-5/group was used due to a limited availability of *Traf3*-deficient bone marrow (kindly sent from Brendan Boyce's lab, Rochester, Minnesota), however the WT and WT + Elig groups were repeated several times including Figure 1 and Figure S1.

In Figure 3D, the mice used are C57BL/6J mice as opposed to the C57BL/KaLwRij used in new Figure 2 and hence BV/TV may vary. There is also a difference in age of the mice at cull due to the way the different in vivo models work. In Figure 2, the experiment commences in 8-week-old female C57BL/KaLwRij and experiment lasts for 23 days (cull and scan the micro-CT at 11 weeks plus 1 day). For Figure 3 using C57BL/6J mice, the experiment started at 4 weeks

of age and were fed on the HFD for 7 weeks prior to 5TGM1 cell injection, the model then runs for another 30 days, which means the mice were sacrificed at 15 weeks plus 2 days (shown in schematic in Figure 3A).

7. The section on the high bone mass diet was hard to follow. The data is interesting but articulation in the results section was hard to follow.

This section has been extensively rewritten (lines 205-241) to aid the clarity of these results and the rationale for using this model.

8. The data on *traf3* ^{-/-} is modest with relatively few animals and limited analysis. It would be good to see a more detailed analysis in the same way is show in figures 1-3.

More detailed analysis of the data shown in Figure 5H has been added to Figure S5G-I.

9. It was unclear whether the patient data is from a single patient. I seem to have missed the supplementary table.

Baseline MM patient characteristics are shown in Supplementary Table 1; nine patient samples were used in total. The OC images shown in Figure 8A and 8C are from one representative experiment whilst the quantifications shown in Figure 8B and 8D are from n=5-7.

10. The authors argue that this agent could be used more broadly. It would be good to understand whether there are potential significant off-target effects anticipated – presumably they would be limited if this is used in other clinical settings.

Eliglustat is used clinically and was FDA approved in 2014 for the treatment of Gaucher's Disease⁷. The most common side effect seen with eliglustat is dyspepsia (heartburn) in approximately 6 out of 100 patients. The most common serious side effect is fainting, in 8 out of every 1,000 patients. The majority of side effects are mild and short-lived⁷.

In addition, the autophagy-dependency of TRAF3 levels has potentially wider clinical applications. In autophagy-deficient A549 cells, elevated TRAF3 levels suppress tumorigenicity⁸, suggesting that eliglustat may have anti-tumour effect in cancers and may treat diseases due to the activation of non-canonical NF- κ B pathway (lines 536-540).

Reviewer #3 (Remarks to the Author):

In this paper the authors reported the involvement of glycosphingolipid in autophagy during osteogenesis. The presented results are potentially interesting and deals with a valuable topic. However, there are some ambiguous parts that need to be organized and make the different parts of the article fluid together and, in addition, there are some English errors that make some parts unclear.

In particular, these are some comments and concerns need to be addressed:

1) lines 40-41 in abstract are not so clear and appear not linked with the first sentence. Generally the abstract is not appealing.

The abstract has been reworded to follow the narrative of the paper more closely and be more accessible to the reader (lines 42-54).

2) line 75, the information about TRAF3 degradation is very general, as the results of this project are focused on autophagy and TRAF3 degradation, it is important to explain it in more detail in both introduction and also discussion sections.

Greater explanation of the process of TRAF3 degradation has been added to the introduction (lines 92-103) and to the discussion (lines 536-540). In addition, the process of autophagy has been clarified further (lines 75-85).

3) line 100, the paragraph title is very general, as the authors have explained lots of results in this section, it is better to write a more detailed title too.

This subheading has been expanded as suggested by the reviewer (line 167).

4) Why did the authors not evaluate Eliglustat on normal control animals to understand the effects of this substance in normal conditions? They have evaluated this drug only on MM group. This must be included.

We previously included this data as Figure S1. This has now been moved to the main figures as Figure 1. We also evaluated eliglustat in both female (Figure 1) and male mice (Figure S1A, B).

5) Why have the female mice been selected for this study while the hormonal cycles can interfere with the experiment and in particular with bone structure? I suggest to evaluate the differences with male mice, if any.

Both male and female mice have been evaluated. As shown in Figure S2, treatment with eliglustat increased BV/TV and other bone indices in males with MM. Furthermore, we also show naive male mice treated with eliglustat (Figure S1A, B).

Supplemental Figure 1

Figure S1.

Supplemental Figure 2

Figure S2.

6) What is the rational explanation for studying the effects of Eliglustat on HFD bone markers? It should be explained.

The section describing the HFD experiment has been rewritten to aid clarity (lines 205-241).

7) What is the role of MG132 on the OC differentiation pathway? In the figure 4 there is information about the role of MG132 on IκBα degradation and also OC precursors have been treated with this substance, but in the text the authors did not indicate its role and why they used it.

The rationale for investigating MG132 has been added to the manuscript in greater detail (lines 316-327).

8) The differentiation of Raw cells into OC is a long process that needs at least 1 week. Why did they treat and evaluate cells only for 1 day (line 264)?

Yes, the reviewer is correct that the formation of RAW264.7 cells into OCs is a long process taking a week or so to occur. The aim of this experiment was to investigate the early steps that are required for the initiation of OC differentiation and hence why the 24-hour timepoint was chosen – sufficient time to allow for TRAF3 changes to occur but short enough to be due to a direct effect of eliglustat.

In addition, our new confocal imaging shows that 15 mins' eliglustat treatment is sufficient to cause TRAF3 accumulation in the RAW264.7 cells (Figure 6E-F) indicating that alterations in TRAF3 are occurring at the early stages of OC differentiation.

9) line 357, it is necessary to indicate “mononuclear cells” isolated from BM instead of “BM” alone that is not clear.

The wording ‘mononuclear cells’ has been added to lines 474 and 477 as well as the figure legend for Figure 8 (lines 490, 492 and 495).

1. Wang C, Hu Q, Shen HM. Pharmacological inhibitors of autophagy as novel cancer therapeutic agents. *Pharmacol Res* **105**, 164-175 (2016).
2. Chicote J, Yuste VJ, Boix J, Ribas J. Cell Death Triggered by the Autophagy Inhibitory Drug 3-Methyladenine in Growing Conditions Proceeds With DNA Damage. *Front Pharmacol* **11**, 580343 (2020).
3. Chen W, *et al.* Autophagy inhibitors 3-MA and LY294002 repress osteoclastogenesis and titanium particle-stimulated osteolysis. *Biomater Sci* **9**, 4922-4935 (2021).
4. Xiu Y, *et al.* Chloroquine reduces osteoclastogenesis in murine osteoporosis by preventing TRAF3 degradation. *J Clin Invest* **124**, 297-310 (2014).
5. Zhu S, *et al.* Bafilomycin A1 Attenuates Osteoclast Acidification and Formation, Accompanied by Increased Levels of SQSTM1/p62 Protein. *J Cell Biochem* **117**, 1464-1470 (2016).
6. Lwin ST, Olechnowicz SW, Fowler JA, Edwards CM. Diet-induced obesity promotes a myeloma-like condition in vivo. *Leukemia* **29**, 507-510 (2015).
7. Mistry PK, *et al.* Effect of oral eliglustat on splenomegaly in patients with Gaucher disease type 1: the ENGAGE randomized clinical trial. *JAMA* **313**, 695-706 (2015).
8. Newman AC, Kemp AJ, Drabsch Y, Behrends C, Wilkinson S. Autophagy acts through TRAF3 and RELB to regulate gene expression via antagonism of SMAD proteins. *Nat Commun* **8**, 1537 (2017).

REVIEWER COMMENTS

Reviewer #1 (Remarks to the Author):

The authors have tried to address all my comments. I am overall satisfied but I think one of key finding is not yet supported by these new data.

Indeed, The increase of TRAF3-LAMP1 co-localization upon Eliglustat it is unclear (new figure 6H). This is a critical point given that the authors have been unable to verify lysosomal accumulation of TRAF3 using biochemical approaches (fig. S6).

The authors should make extra efforts to convincingly show that TRAF3 molecules can be seen in the lysosomes (or in autophagosomes)

Reviewer #3 (Remarks to the Author):

Authors responded to all of my comments clearly and made changes in the manuscript precisely.

Reviewer #4 (Remarks to the Author):

The Authors responded to all the questions of the reviewer2# but not to the question 6.

I suggest to repeat mice experiments with a low number of mice as requested by the reviewer two.

Dear Editor,

We are very grateful for the helpful and constructive reviewers' comments. The authors hope that our response to the suggested amendments will meet your requirement for publication.

Reviewer 1

The authors have tried to address all my comments. I am overall satisfied but I think one of key finding is not yet supported by these new data. Indeed, the increase of TRAF3-LAMP1 co-localization upon Eliglustat it is unclear (new figure 6H). This is a critical point given that the authors have been unable to verify lysosomal accumulation of TRAF3 using biochemical approaches (fig. S6). The authors should make extra efforts to convincingly show that TRAF3 molecules can be seen in the lysosomes (or in autophagosomes)

To further address the reviewer's concern, we used a lysosome isolation kit (LYSISO1, Sigma-Aldrich) to enrich the lysosomes from RAW264.7 cells. As shown below, there was a significant enrichment of LAMP1 compared to the whole cell lysate following the lysosomal isolation kit (Rebuttal Figure 1).

Rebuttal Figure 1. LAMP1 is enriched in the lysosome layer. RAW264.7 cells were lysed in a Dounce homogenizer and centrifuged at 1,000×g for 10 minutes according to the lysosome isolation kit protocol to create the whole cytosolic lysate. The remaining supernatant was centrifuged at 20,000×g for 20 minutes and the pellet collected for ultracentrifuge at 150,000×g for 4 hours, after which the lysosome layer was collected. Equal amounts of the cytosolic lysate and lysosome layer were loaded for Western blot from Control (Ctr, DMSO), CQ, BafA1 and Elig groups.

Following treatment with BafA1, CQ and eliglustat, there was a significant increase in TRAF3 protein in lysosome-enriched layer as evidenced by Western blot (Rebuttal Figure 2A-B). This has been added to the main text of the manuscript as Figure 6M and 6N.

In addition, similarly to Newman *et al.* Nat Comms 2017 (Supplementary Fig. 3c)¹ and by Xiu *et al.* JCI 2014 (Figure 2F)², we used LAMP2 (ab13524, abcam) to repeat the confocal experiment to see if TRAF3 colocalized with the lysosomes. According to the quantified data, there was a significant increase of TRAF3 and LAMP2 colocalization after treatment with BafA1, CQ and eliglustat (Rebuttal Figure 2C-E). This has been added to the main text of the manuscript as Figure 6O and 6P (60minutes), and Supplemental Figure 6C and 6D (120minutes).

Rebuttal Figure 2. Eliglustat accumulates TRAF3 in lysosome in RAW264.7 cells.

(A) The separated lysosome layers from control group (DMSO), BafA1 (10 nM) group (2 hours treatment), CQ (20 µM) group (12 hours treatment) and eliglustat (50 µM) group (12 hours treatment) were blotted with TRAF3 and LAMP1. (B) TRAF3 protein level was quantified relative to the lysosome marker LAMP1. n=12. (C-E) Confocal images of TRAF3 and LAMP2 in RAW264.7 cells treated with BafA1, CQ, and eliglustat. Scale bar represents 10µm (C). The percentage of TRAF3 and LAMP2 colocalization were quantified at 60 minutes (D) and 120mins (E), n≥4. Error bars correspond to SEM. **P<0.01, ***P<0.001, ****P<0.0001. Statistical analysis was performed using One-way ANOVA.

Reviewer 3

Authors responded to all of my comments clearly and made changes in the manuscript precisely.

Thank you.

Reviewer 4 (Remarks to the Author):

The Authors responded to all the questions of the reviewer2# but not to the question 6. I suggest to repeat mice experiments with a low number of mice as requested by the reviewer two.

In accordance with the reviewer’s suggestion, the TRAF3 BM chimera experiment was repeated using a further 5-6 mice per group. As previously described in the main Figure 5N, CD45.1 mice were lethally irradiated and transplanted with CD45.2 BM cells from TRAF3 KO mice or BM cells from control littermates. After 6 weeks, reconstitution with CD45.2 cells in the CD45.1 mice was assessed by FACS analysis of myeloid cells derived from PBMC (Rebuttal Figure 3A). Mice were then treated with eliglustat for 19 days and the tibiae were harvest for micro-CT analysis (Rebuttal Figure 3B). Consistent with previous findings, eliglustat was unable to increase the bone volume in recipient mice transplanted with TRAF3 KO BM cells underscoring the importance of blocking TRAF3 degradation for the effect of eliglustat on osteoclastogenesis (Rebuttal Figure 3C). Due to between experiment variation in engraftment and in the absolute values for BV/TV%, the data cannot be combined with that in Figure 5N thus this new data has been included for rebuttal purposes only. Data shown in Figure 5N is now described as one experiment representative of two independent experiments with n=4-6 for each group.

Rebuttal Figure 3. TRAF3 KO BM chimera recipient mice resistant to eliglustat-induced bone volume increase. (A) Lethally irradiated recipient CD45.1 mice were reconstituted with littermate (WT) or myeloid specific TRAF3 knockout (LysM-Cre+,

Traf3 fl/fl) CD45.2 BM cells for 6 weeks and the reconstitution efficacy was verified by flow cytometry of the PBMC. **(B)** Micro-CT reconstruction images of WT, WT+Elig, traf3 KO and traf3 KO+Elig mice (eliglustat treated for 19 days, 150 mg/kg/day). **(C)** bone volume over total volume (BV/TV) of tibiae was quantified by micro-CT (n=5-6 per group). Data represented as mean \pm SEM. **P<0.01, ***P<0.001, ns means non-significant. Statistical analysis was performed using One-way ANOVA.

References

1. Newman AC, Kemp AJ, Drabsch Y, Behrends C, Wilkinson S. Autophagy acts through TRAF3 and RELB to regulate gene expression via antagonism of SMAD proteins. *Nat Commun* **8**, 1537 (2017).
2. Xiu Y, *et al.* Chloroquine reduces osteoclastogenesis in murine osteoporosis by preventing TRAF3 degradation. *J Clin Invest* **124**, 297-310 (2014).

REVIEWERS' COMMENTS

Reviewer #1 (Remarks to the Author):

I have no additional comments.

Reviewer #4 (Remarks to the Author):

Authors responded adequately to the reviewer's requests.

Dear Editor,

We are very glad that our previous point-to-point reply addressed all reviewers' comments. Hope this version of the manuscript will meet your requirements for publication.

Reviewer #1 (Remarks to the Author):

I have no additional comments.

Thank you.

Reviewer #4 (Remarks to the Author):

Authors responded adequately to the reviewer's requests.

Thank you.